# Single-pass Adaptive Image Tokenization for Minimum Program Search

**Shivam Duggal**    **Sanghyun Byun**[†]    **William T. Freeman**    **Antonio Torralba**    **Phillip Isola**

Massachusetts Institute of Technology    LG Electronics[†]

## Abstract

According to Algorithmic Information Theory (AIT) – Intelligent representations compress data into the shortest possible program that can reconstruct its content, exhibiting low Kolmogorov Complexity (KC). In contrast, most visual representation learning systems use fixed-length representations for all inputs, ignoring variations in complexity or familiarity. Recent adaptive tokenization methods address this by allocating variable-length representations but typically require test-time search over multiple encodings to find the most predictive one. Inspired by Kolmogorov Complexity principles, we propose a single-pass adaptive tokenizer, KARL, which predicts the appropriate number of tokens for an image in a single forward pass, halting once its *approximate* KC is reached. The token count serves as a proxy for the minimum description length. KARL's training procedure closely resembles the Upside-Down Reinforcement Learning paradigm, as it learns to conditionally predict token halting based on a desired reconstruction quality. KARL matches the performance of recent adaptive tokenizers while operating in a single pass. We present scaling laws for KARL, analyzing the role of encoder/decoder size, continuous vs. discrete tokenization and more. Additionally, we offer a conceptual study drawing an analogy between Adaptive Image Tokenization and Algorithmic Information Theory, examining the predicted image complexity (KC) across axes such as structure vs. noise and in- vs. out-of-distribution familiarity – revealing alignment with human intuition.

**Code:** https://github.com/ShivamDuggal4/kolmogorov-tokenizer.
**Keywords:** Representation Learning, Adaptive Tokenization, Compression, Algorithmic Information Theory, Kolmogorov Complexity, Upside-Down RL.

## 1   Introduction

Compression lies at the heart of intelligence [1–3]. The universal intelligence framework from Legg and Hutter [4], inspired from the Solomonoff Prior, formalizes *general intelligence* as an agent's expected performance across a wide range of computable tasks or environments, each weighted by its simplicity—more precisely, by its **Kolmogorov complexity** [5]. Agents are thus rewarded for performing well on environments that are easier to describe and penalized less for failures on highly complex ones. The core principle behind this framework is a formalization of **Occam's razor** [6] — simpler explanations are inherently more plausible and tend to generalize better, making them more desirable to optimize. This simplicity bias underlies much of machine learning—ranging from classical dimensionality reduction [7, 8] to modern representation learning [9] — all of which aim to capture the underlying regularities in data using as few components as possible. In this view, intelligent data representations capture the structure of their input in the simplest, most compressed form—ideally with minimal Kolmogorov Complexity (KC) or its practical counterpart, Minimum Description Length (MDL)—while preserving the information necessary for downstream tasks, effectively performing task-aware lossless compression.

39th Conference on Neural Information Processing Systems (NeurIPS 2025).

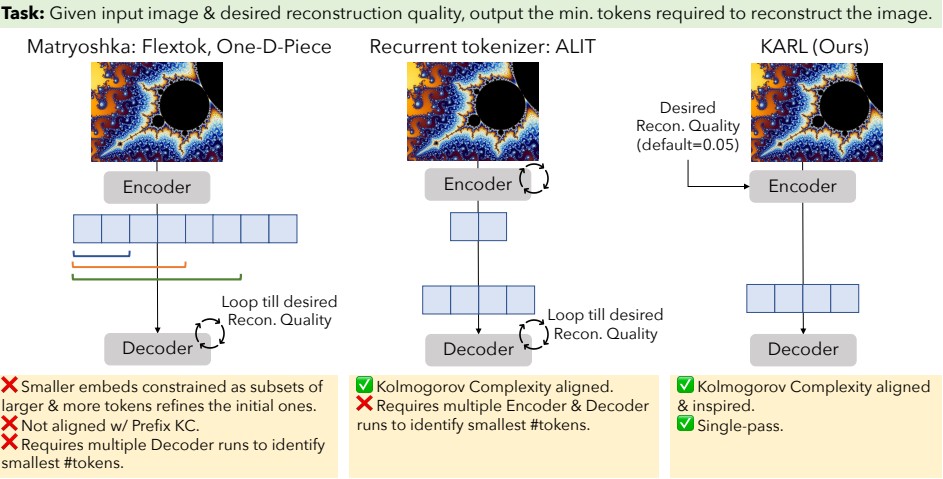

Figure 1: **Comparing adaptive tokenizers:** Matryoshka-style methods constrain smaller token sets to be subsets of larger ones and require multiple decoder passes. Recurrent approaches like ALIT perform iterative encoder-decoder loops to meet reconstruction quality. In contrast, KARL employs a *single-pass encoder guided by Kolmogorov Complexity, avoiding iterative refinement.* KARL's encoder outputs both token embeddings & halting probabilities; the non-halted tokens constitute the minimal program sufficient to reconstruct the image to the target quality using the decoder.

In this work, we focus on visual representation learning from the perspective of *maximum compression*. Most existing representation learning algorithms—such as SimCLR [10], DINO[11], MAE [12], VAE [13], VQGAN [14] — employ objectives like contrastive learning, self-distillation, or reconstruction to map input images into fixed-length embeddings. While these methods have proven effective for downstream tasks such as classification, VQA and image generation, they produce representations of *uniform length* for all images, regardless of image complexity, familiarity, or task relevance. This uniformity contradicts the principles of Kolmogorov Complexity, which defines the complexity of an object as the shortest program to generate it.

Recently, there has been growing interest in adaptive tokenization, which assign variable-length representations to different images at test time. These approaches are more aligned with the principle of Occam's razor and KC, as they encode each input with a variable number of tokens required to preserve essential information. They also resonate with Epicurus's principle [15], which encourages retaining multiple possible explanations — **Adaptive tokenization** can thus be seen as dynamically selecting among multiple candidate representations depending on image complexity or familiarity.

Among adaptive visual tokenizers, two broad families have emerged: Matryoshka-style [16] or prefix-based models (or tail-dropping representations) [17, 18], and recurrent or iterative tokenizers [19], which resemble Turing machines in their step-wise construction of a representation. Fig. 1 compares these tokenizers. The first category learns a large fixed-length embedding for each image, designed so that any prefix—or subset—can serve as a valid explanation. While this offers a mechanism to approximate the minimum required representation length, it assumes that all explanations are nested within a single, maximal one. This contradicts a core property of *prefix Kolmogorov complexity* [5, 20], which states that the shortest valid description may not be a subset of longer ones. Such nesting can thus impose a strong inductive bias that limits alignment with the principles of minimality and intelligent behavior – any additional token utilized for an image has the role of *refining the existing representation, rather than enabling the discovery of a potentially more optimal solution* under increased memory constraints. In contrast to Matryoshka-style approaches, recurrent or iterative tokenizers *progressively allocate more tokens or memory over multiple network iterations* to represent an image, typically halting once a target reconstruction loss is reached. Methods such as ALIT [19] and ElasticTok [21] fall into this category. Although these recurrent approaches are more faithful to the principles of Kolmogorov Complexity, both categories of approaches incur a significant practical cost when adapting representation lengths to input images — identifying minimal embedding requires repeatedly rolling out the encoder-decoder pipeline, which limits their efficiency in real-world applications like VLMs & video models, demanding fast and lightweight inference.

Contrary to prior approaches that rely on iterative search to generate maximally compressed representations at test time, we propose **KARL — Kolmogorov-Approximating Representation Learning**,

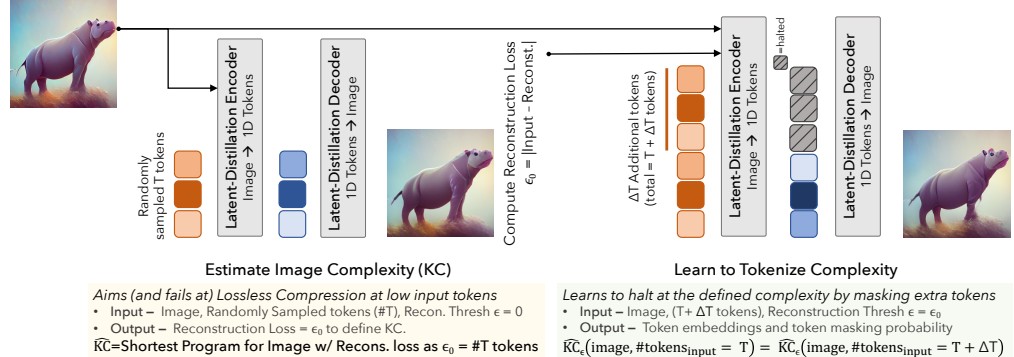

**Estimate Image Complexity (KC)**

*Aims (and fails at) Lossless Compression at low input tokens*
- **Input** – Image, Randomly Sampled tokens (#T), Recon. Thresh $\epsilon = 0$
- **Output** – Reconstruction Loss $= \epsilon_0$ to define KC.

$\widehat{KC}$=Shortest Program for Image w/ Recons. loss as $\epsilon_0$ = #T tokens

**Learn to Tokenize Complexity**

*Learns to halt at the defined complexity by masking extra tokens*
- **Input** – Image, (T+ $\Delta$T tokens), Reconstruction Thresh $\epsilon = \epsilon_0$
- **Output** – Token embeddings and token masking probability

$\widehat{KC}_\epsilon(\text{image}, \#\text{tokens}_{\text{input}} = T) = \widehat{KC}_\epsilon(\text{image}, \#\text{tokens}_{\text{input}} = T + \Delta T)$

Figure 2: **KARL** – a one-pass adaptive tokenizer inspired from the principles of KC in being able to use only the sufficient token count for an image, halting any extra token beyond the image's complexity. The training procedure operates by first attempting lossless compression and later using the failed reconstruction loss measures as guiding signal to halt any extra token than provided during the lossless compression attempt. Such a training strategy optimizes the network for reconstruction tasks closer to an image's inherent complexity, implicitly guiding it to allocate tokens adaptively.

a novel method for variably-compressed visual representation learning in a single forward pass. Inspired by the principle of Kolmogorov Complexity (KC), KARL learns to approximate the minimal number of tokens required to reconstruct an image up to a target reconstruction loss. To achieve this, we introduce a *loss-conditioned supervision strategy* that trains the model to identify and mask unnecessary tokens in a fully differentiable manner. Each training iteration follows a two-phase process resembling Schmidhuber [22]'s upside-down reinforcement learning (RL) paradigm: first, the model estimates image complexity by attempting near-lossless reconstruction under a randomly sampled token budget; the resulting reconstruction loss then serves as the task condition for the second phase, where the model learns to achieve the same quality using more tokens while halting the surplus ones. This setup enables the model to *adaptively allocate tokens in a single forward pass* by treating the reconstruction loss as a task-specifying reward—akin to upside-down RL. At inference, given an image and a desired reconstruction quality, KARL's encoder predicts both token embeddings and halting probabilities. Tokens with high halting probability are excluded from decoding, yielding an efficient, adaptive representation. In effect, KARL *implicitly searches for the shortest sufficient program*—outputting only the necessary tokens to match the target quality (see Tab. 2).

We compare KARL both quantitatively (Tab. 1, Tab. 2) and qualitatively (Fig. 4, Fig. 15, Fig. 16) against recent adaptive tokenization methods using standard image reconstruction metrics. KARL performs competitively across all single-image metrics — LPIPS, SSIM, PSNR, and DreamSim. We also compare these tokenizers in terms of the minimum number of tokens ($\acute{KC}$) required to reconstruct the input image to a given reconstruction quality (Tab. 2, Fig. 17, Fig. 18).

Beyond introducing **KARL as a one-step adaptive tokenizer**, we draw connections between adaptive image tokenization and the framework of *Algorithmic Information Theory (AIT)* [15]. In particular, the minimal token count predicted by the adaptive tokenizer can be considered as an approximation of the *incomputable* notion of Kolmogorov Complexity (KC). We perform both quantitative and qualitative analyses of KARL's approximation of KC through Fig. 5, Fig. 7, Fig. 17, and Fig. 18. Additionally, KARL's complexity estimates—measured by the number of latent tokens used—show strong alignment with human annotations of visual complexity (Fig. 14). Beyond Kolmogorov Complexity, we suggest that adaptive tokenizers may also capture higher-order AIT concepts such as *Sophistication* and *Logical Depth*, though this remains a promising direction for future work. Overall, our work positions adaptive tokenization, KARL in particular, as a means of *learning intelligent representations: maximally compressed yet predictive, and in alignment wtih the principles of AIT*.

## 2 Related Work

A central goal in representation learning and tokenization is *dimensionality reduction*—mapping high-dimensional inputs to lower-dimensional but structured manifolds. These compact representations are believed to generalize better and are widely used to train more efficient and effective neural networks. As a result, compressed visual learning is increasingly important in the current landscape of AI.

Focusing primarily on reconstruction-based tokenizers, the predominant approaches include VAE [13] and VQGAN-style [14] models. Compression in these architectures is typically achieved

---

**Input & Preprocessing:** Input image **img**; token budget $T$; desired loss $\mathcal{L} = 0$;
Convert **img** to `img_tokens` $\in \mathbb{R}^{B \times HW \times D}$

---

**Estimating image complexity by aiming Lossless Compression under budget T**
1: // Step 1: Token Initialization
   Initialize $\mathbf{z}_0 \in \mathbb{R}^{B \times T \times D}$ as `init_tokens`
2: // Step 2: Initial Encoding with Zero Epsilon (Best possible reconstruction under budget T)
   $\mathbf{z}_{\text{enc}, \_} \leftarrow \texttt{Encoder}(\mathbf{x}, \mathbf{z}_0, \epsilon = 0); \quad \hat{\mathbf{img}}_0 \leftarrow \texttt{Decoder}(\mathbf{z}_{\text{enc}})$
3: // Step 3: Encoder maps image $\rightarrow$ T tokens. Decoder transforms T tokens $\rightarrow$ image.
   Compute Reconstruction Error
   $\epsilon_0 \leftarrow \left| \hat{\mathbf{img}}_0 - \mathbf{img} \right|$

---

**Learning to tokenize complexity by masking additional tokens:** Increased token budget $T + \Delta T$
4: // Step 4: Token Re-Initialization
   Extend the token budget: $\mathbf{z}_1 \leftarrow \texttt{append\_tokens}; \quad \mathbf{z}_1 \in \mathbb{R}^{B \times (T + \Delta T) \times D}$
5: // Step 5: Use the minimum required token count to achieve $\epsilon_0$; $\hat{\mathbf{K}}\mathbf{C} = \mathbf{T}$ by design
   $\mathbf{z}_{\text{final}}, \omega_{\text{final}} \leftarrow \texttt{Encoder}(\mathbf{x}, \mathbf{z}_1, \epsilon = \epsilon_0)$
6: // Step 6: Sample learned token embedding with halting probability, $\omega_{\text{final}} < 0.75$
   $\mathbf{z}_{\text{minimal}} \leftarrow \texttt{Index } \mathbf{z}_{\text{final}} \forall \; (\omega_{\text{final}} < 0.75)$
7: $\hat{\mathbf{img}} \leftarrow \texttt{Decoder}(\mathbf{z}_{\text{minimal}})$
8: // Step 8: Optimize using Reconstruction Losses & Cross-Entropy on $\omega_{final}$
   Halting probability $\omega_{\text{final}} = 1$ for last $\Delta T$ tokens (extra budget), 0 for the first $T$ tokens.

---

Figure 3: **Training algorithm.** KARL follows an upside-down RL paradigm—(a) **Estimate Image Complexity** stage samples task-defining inputs {image, token budget T, reconstruction error $\epsilon_0$} by attempting lossless compression ($\epsilon = 0$). (b) **Learn to Tokenize** stage trains the model—conditioned on $\epsilon_0$—to match the same quality using T+$\Delta$T tokens while halting the extra (rightmost) $\Delta$T.

through spatial down-sampling in convolution-based encoders or via fixed patch-to-token mappings as in vision transformers [23] — after which no further compression occurs and the initial token count is preserved. This rigid patch-token binding limits the ability to compress the input image effectively. The absence of token-level compression may pose challenges for scaling to large vision tasks, such as video generation and understanding. A widely accepted alternative to patch-token binding is Perceiver's [24, 25] 1D tokenization strategy, which maps all inputs to a fixed-length 1D sequence—enabling modality-agnostic compression. Recent works like Titok [26] and FlowMo [27] follow this paradigm. While these methods achieve better compression than 2D patch-based systems, they allocate a fixed token capacity to all images.

Intelligent representation learning should aim for maximum compression conditioned on the input and task. Adaptive tokenization approaches [17–19, 21, 28] address this by training tokenizers to map images to a variable number of tokens, enabling per-image adaptive inference. Among these, Matryoshka-style models such as FlexTok [17], One-D-Piece [18], and ElasticTok [21] are prominent—they assume that any subset of a full embedding should remain representative. However, as discussed earlier, deriving compressed representations as strict subsets of longer ones may not yield optimal token allocations. An alternative strategy is iterative adaptive tokenization, where tokens (or memory) are incrementally allocated over multiple steps. ALIT [19], for example, introduces an RNN-based Transformer for adaptive-length image tokenization.

Both category of methods, however, require iterative inference involving multiple encoder-decoder passes to assess image complexity and assign representations accordingly. A recent work, CAT [29], instead leverages an LLM to predict image complexity before selecting the appropriate nested VAE level for compression. In contrast, we propose a single-pass adaptive tokenization strategy that implicitly learns to sample only the required token count per image—offering an efficient mechanism to estimate input complexity via the approximated minimal program size. Furthermore, we draw connections between adaptive image tokenization and foundational concepts in Algorithmic Information Theory (AIT) [15], *providing a fresh perspective on visual learning.*

## 3 Kolmogorov-Approximating Representation Learning

Representation learning seeks to map input observations to compact, meaningful embeddings that retain essential information for downstream tasks. Our goal is to learn **intelligent data representations** that are both highly compressed & predictive. We formalize this as a dual optimization problem:

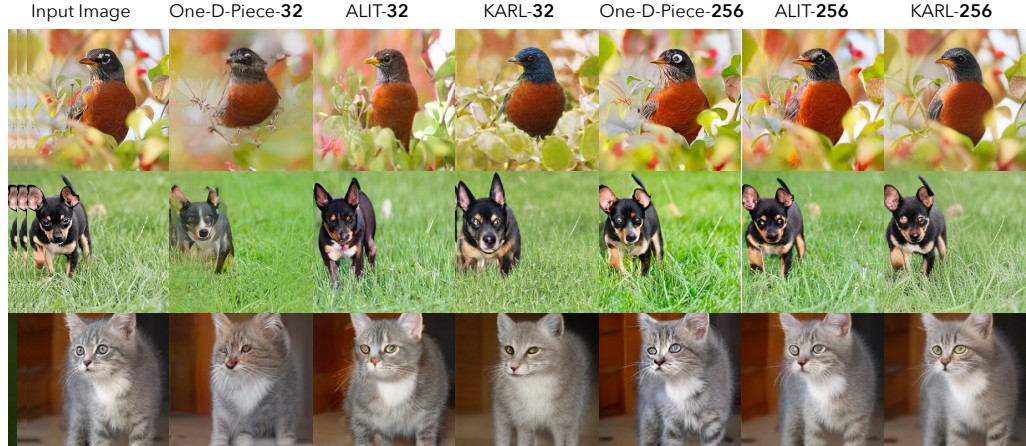

| Input Image | One-D-Piece-**32** | ALIT-**32** | KARL-**32** | One-D-Piece-**256** | ALIT-**256** | KARL-**256** |

Figure 4: **Reconstruction Analysis on ImageNet100:** KARL and ALIT produce higher-quality reconstructions than One-D-Piece, with KARL achieving strong single-image metrics. One-D-Piece consistently generates blurry images at low token counts satisfying FID metric more.

  (i) minimizing reconstruction loss to ensure the latent representation faithfully captures the input

  (ii) minimizing the representation length to achieve maximal compression.

In this work, we focus exclusively on reconstruction-based encoder-decoder approaches (image $\rightarrow$ embedding or tokens $\rightarrow$ image), as they naturally align with the goal of lossless compression. Moreover, since fixed-length representations violate the core principles of Kolmogorov Complexity (KC), this work centers on variable-length representation learning through adaptive tokenization. Recent adaptive tokenizers rely on iterative encoder-decoder runs at inference time to identify minimum tokens to reconstruct an image up to a desired quality. In contrast, we pose the following question: Given a token budget, *can we determine in a single shot how many tokens are sufficient to form an intelligent representation—one that is both maximally compressed and predictive?*

Before diving deeper into this question, it's important to acknowledge two fundamental challenges: both identifying *the smallest representation length and perfect lossless compression in deep learning models are practically unattainable*. We leverage these limitations as a tool in designing tokenizer to perform minimum program search [1]. We propose a loss-conditioned adaptive tokenization approach, dubbed **Kolmogorov-Approximating Representation Learning (KARL)**, which learns to approximate the minimum description length of an image—defined as the minimal number of tokens needed—subject to a given reconstruction threshold. More formally, KARL is an encoder-decoder architecture that, at test time, takes an image, a maximum token budget, and a satisfiable reconstruction loss threshold as input, and outputs only the necessary tokens (i.e., the approximated minimum program size), while masking out the rest. The use of loss-conditioning with variable reconstruction thresholds can also be interpreted as prompting the model with alternative tasks, since different downstream applications in computer vision require different levels of detail to be preserved. *KARL produces variably compressed representations based on the target loss magnitude—allocating more tokens for harder tasks—aligning with the spirit of Universal Intelligence framework [4].*

**How do we train KARL?** We start by outlining the core encoder-decoder architecture used for adaptive tokenization, which maps images to 1D latent tokens. Building on this, we introduce our loss-conditioned tokenization strategy, which enables training of the proposed tokenizer that learns to approximate the minimal tokens necessary to satisfy a given reconstruction threshold.

**1D Tokenization:** Like most recent adaptive tokenizers and inspired by the Perceiver model [24], KARL learns to distill an input image into 1D latent tokens that are not constrained to fixed spatial patches. We first map the image to a grid of 2D latent tokens using a pretrained VAE / VQGAN, resulting in 256 tokens for a $256 \times 256$ image. These 2D tokens are concatenated with a set of initialized 1D latent tokens along the token dimension and passed to a latent distillation encoder—a transformer-based encoder that performs full self-attention across all tokens. The goal of the encoder is to distill the 2D image tokens into a variable-length 1D representation. For reconstruction, a decoder with the same architecture performs cross-attention between the masked 2D token grid

---

[1]We use — minimum program size, MDL, minimum tokens or approximated KC or $\hat{\text{KC}}$ interchangeably.

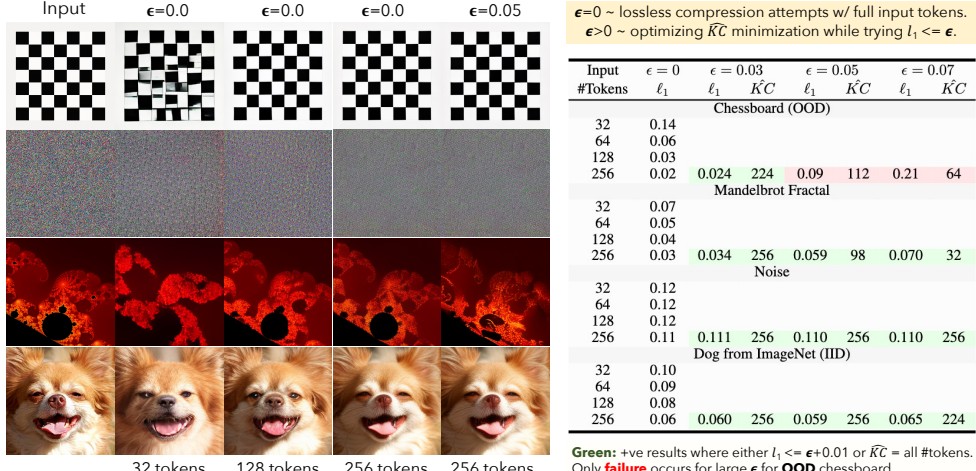

| Input | ε=0.0 | ε=0.0 | ε=0.0 | ε=0.05 |
|---|---|---|---|---|

ε=0 ~ lossless compression attempts w/ full input tokens.
ε>0 ~ optimizing $\hat{KC}$ minimization while trying $\ell_1$ <= ε.

| Input #Tokens | ε = 0 $\ell_1$ | ε = 0.03 $\ell_1$ | ε = 0.03 $\hat{KC}$ | ε = 0.05 $\ell_1$ | ε = 0.05 $\hat{KC}$ | ε = 0.07 $\ell_1$ | ε = 0.07 $\hat{KC}$ |
|---|---|---|---|---|---|---|---|
| \multicolumn{8}{c}{Chessboard (OOD)} | | | | | | | |
| 32 | 0.14 | | | | | | |
| 64 | 0.06 | | | | | | |
| 128 | 0.03 | | | | | | |
| 256 | 0.02 | 0.024 | 224 | 0.09 | 112 | 0.21 | 64 |
| \multicolumn{8}{c}{Mandelbrot Fractal} | | | | | | | |
| 32 | 0.07 | | | | | | |
| 64 | 0.05 | | | | | | |
| 128 | 0.04 | | | | | | |
| 256 | 0.03 | 0.034 | 256 | 0.059 | 98 | 0.070 | 32 |
| \multicolumn{8}{c}{Noise} | | | | | | | |
| 32 | 0.12 | | | | | | |
| 64 | 0.12 | | | | | | |
| 128 | 0.12 | | | | | | |
| 256 | 0.11 | 0.111 | 256 | 0.110 | 256 | 0.110 | 256 |
| \multicolumn{8}{c}{Dog from ImageNet (IID)} | | | | | | | |
| 32 | 0.10 | | | | | | |
| 64 | 0.09 | | | | | | |
| 128 | 0.08 | | | | | | |
| 256 | 0.06 | 0.060 | 256 | 0.059 | 256 | 0.065 | 224 |

**Green:** +ve results where either $\ell_1$ <= ε+0.01 or $\hat{KC}$ = all #tokens.
Only **failure** occurs for large ε for **OOD** chessboard.

| 32 tokens | 128 tokens | 256 tokens | 256 tokens |
|---|---|---|---|

Figure 5: **Estimating $\hat{KC}$ across structured, noisy, and OOD inputs.** Recon. quality and $\hat{KC}$ vary by image type and token count. The dog (IID) compresses well; the OOD chessboard does not. High Noise and high structure both yield high $\hat{KC}$ (like theoretical KC), but adaptive tokenizers can distinguish: for noise, increasing tokens doesn't improve $\ell_1$ i.e. $\Delta = \ell_1(256 \text{ tokens}) - \ell_1(32 \text{ tokens}) \approx 0$. For redundant chessboard structure, $\Delta$ turns to be too high – meaning what's actually *interesting* lies in the mid $\Delta$ range – suggesting potential to model AIT notions like *Sophistication, Logical Depth*. Table on the right shows how well KARL minimizes $\hat{KC}$ under the constraint $\ell_1 \leq \epsilon$ for diff. input $\epsilon$.

(representing positional structure of the original 2D image tokens) and the encoded 1D latent tokens, allowing the model to reconstruct the image via predicted 2D tokens and VQGAN or VAE decoder.

With the goal of training a one-shot adaptive tokenizer, the encoder is tasked with not only predicting the embedding of each 1D token, but also estimating a halting probability for each token. If a token's halting probability exceeds a given threshold, it is considered inactive—excluded from decoding—and thus does not contribute to the shortest program of the image. However, we do not have direct supervision for these halting probabilities. Furthermore, learning when to halt or mask using only a reconstruction objective is non-trivial: masking is non-differentiable, and common techniques like reinforcement learning or straight-through estimation [30] are highly unstable, especially alongside latent quantization [14]. *How do we train the model to mask out tokens, enabling one-shot prediction of sufficient token count?* We address this by framing the problem as an upside-down RL formulation [22] and introducing a loss-conditioned training strategy inspired by the principles of Kolmogorov Complexity [5]. Fig. 9, Fig. 3 showcase the computational graph & the training algorithm.

**Loss-Conditioned Training of KARL:** We define the approximate Kolmogorov Complexity ($\hat{KC}$) of an image $x$ under a reconstruction threshold $\epsilon$ and a maximum given token budget, $T$ as:
$$\hat{KC}_\epsilon(x, T) = \min \{t \leq T \mid \mathcal{L}_{\text{rec}}(x, \hat{x}_t) \leq \epsilon\}$$

where $\hat{x}_t$ is the reconstructed image using only $t$ of the available $T$ representation tokens, and $\mathcal{L}_{\text{rec}}$ is the reconstruction error (pixel-wise error in our setup). In essence, $\hat{KC}_\epsilon(x, T)$ denotes the smallest token count sufficient to reconstruct the image within the given error bound $\epsilon$, while respecting the token budget $T$. This provides a formal, task-specific interpretation of KC that inspired our training.

Now, to encourage the network to use (fewer) $t < T$ and successfully mask out the extra $(T - t)$ tokens, we set up the training procedure inspired by the following KC invariance property:
$$\hat{KC}_\epsilon(x, T) = \hat{KC}_\epsilon(x, T + \Delta T)$$

i.e., increasing the input token budget should not change the KC, an inherent property of the image. This ensures that once the model has learned the min. sufficient number of tokens $t$ to reconstruct the image within the given error threshold ($\epsilon$), adding more tokens is not required to repeat the same task.

**Training Procedure:** To train KARL, each training iteration involves two runs of the encoder-decoder pipeline, reflecting the principle that *Kolmogorov Complexity (KC) remains invariant once the model has identified a sufficient token count to reconstruct the input*. In the early 2000s, Vereshchagin and Vitanyi [31] highlighted the role of Kolmogorov Complexity, particularly the **Kolmogorov Structure Function**, in lossy compression (see their Example II.4), providing support for our proposed approach.

**1. Estimating Image Complexity:** Given an input image $\mathbf{x}$, the model first attempts near-lossless compression using a given token budget $T$. It is trained to minimize a standard objective that combines reconstruction and latent quantization losses:

$$\mathcal{L}_{\text{EIC}} = \mathcal{L}_{\text{recon}}(\hat{\mathbf{x}}_T, \mathbf{x}) + \beta \, \mathcal{L}_{\text{quant}}(\mathbf{z}_T)$$

where $\hat{\mathbf{x}}_T = \texttt{Decoder}(\texttt{Encoder}(\mathbf{x}, T, \epsilon = 0))$, and $\beta$ balances the quantization loss. The conditioning parameter $\epsilon = 0$ directs the encoder to aim for perfect reconstruction. Let the resulting empirical reconstruction error be recorded as $\epsilon_0 = |\hat{\mathbf{x}}_T - \mathbf{x}|$; this serves as the target reconstruction quality for the second run. By randomly sampling the input budget $T$ in each iteration, we simulate triplets {image, estimated complexity $T$, reconstruction error $\epsilon_0$} for all images in the batch, which form the basis for learning to tokenize by complexity. Fig. 6 showcases a representative batch output from the Estimate Image Complexity (EIC) phase, which serves as input to the subsequent Learning to Tokenize (LTC) phase within the same training iteration.

**Note:** $\epsilon_0$ and $\mathcal{L}_{\text{recon}}$ need not be the same. In most experiments, the loss used to define the reconstruction condition ($\epsilon_0$) is the image-level $\ell_1$ loss, whereas $\mathcal{L}_{\text{recon}}$—used to compute training gradients—can vary: it may be a loss over reconstructed 2D tokens, image-level $\ell_1$, or a learned perceptual or adversarial loss. For exact formulations of $\mathcal{L}_{\text{recon}}$, see Appendix A.2.

**2. Learning to Tokenize at Estimated Complexity:** In the second run, we take the generated simulated triplet of image, complexity $T$, and predicted error $\epsilon_0$, and increase the encoder's input token budget to $T + \Delta T$. The encoder, conditioned on $\epsilon_0$, predicts both the token embeddings & halting probabilities $\boldsymbol{\omega} \in [0, 1]^{T+\Delta T}$, indicating which tokens are essential for reconstruction. To encourage the model to disregard additional $\Delta T$ tokens, we optimize the loss:

$$\mathcal{L}_{\text{LTC}} = \mathcal{L}_{\text{recon}}(\hat{\mathbf{x}}_{T+\Delta T}, \mathbf{x}) + \beta \, \mathcal{L}_{\text{quant}}(\mathbf{z}_{T+\Delta T}) + \lambda \, \mathcal{L}_{\text{halt}}(\boldsymbol{\omega})$$

where $\hat{\mathbf{x}}_{T+\Delta T} = \texttt{Decoder}(\texttt{Encoder}(\mathbf{x}, T + \Delta T, \epsilon = \epsilon_0))$. The encoder is conditioned on the target error $\epsilon_0$ obtained from the first run, encouraging it to match the same reconstruction quality while identifying which tokens are unnecessary. During decoding, halted tokens are excluded from the self-attention operation and therefore do not contribute to the minimal program. The halting loss is defined as:

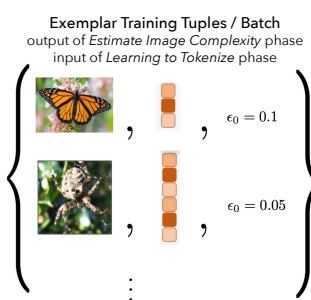

Exemplar Training Tuples / Batch
output of *Estimate Image Complexity* phase
input of *Learning to Tokenize* phase

$\epsilon_0 = 0.1$

$\epsilon_0 = 0.05$

Figure 6: EIC phase provides the task-conditioning (T and $\epsilon_0$) for the second LTC phase.

$$\mathcal{L}_{\text{halt}}(\boldsymbol{\omega}) = \text{BCE}(\boldsymbol{\omega}_{0:T}, \mathbf{0}) + \text{BCE}(\boldsymbol{\omega}_{T:T+\Delta T}, \mathbf{1})$$

encouraging the model to assign low halting probability to essential tokens ($\omega_i \approx 0$ for $i < T$) and high halting probability to redundant ones ($\omega_i \approx 1$ for $i \geq T$). The total training loss for each iteration is $\mathcal{L}_{\text{EIC}} + \mathcal{L}_{\text{LTC}}$. The overall training procedure parallels Upside-Down RL, where the first phase generates a task-defining condition—reconstruction quality—analogous to how Schmidhuber [22] reformulate RL as supervised learning by treating rewards as input conditions. At inference, *only the second adaptive tokenization run is executed*: given a maximum token budget and a target recon. threshold $\epsilon$ (default = 0.05), KARL predicts both token embeddings & halting prob. in a single pass.

**This training reveals an important phenomenon:** The procedure induces a self-supervised curriculum in which the network is only tasked with compressing data to the extent actually feasible—relative to the true complexity of the image. Consider highly complex images, which cannot achieve low error with a small token budget: these are *never paired with a (low-loss, low-token-count) condition* during the second training run. As a result, the training signal is either a strict zero-loss target (for near-perfect reconstruction) or a threshold derived from a failed attempt at lossless compression. In both cases, *the network is optimized for reconstruction tasks that match the image's intrinsic complexity*, implicitly guiding it to allocate tokens adaptively.

## 4 Adaptive Image Tokenization meets Algorithmic Information Theory

From the perspective of representation learning, several seminal frameworks—such as *Algorithmic Information Theory (AIT)*, Solomonoff Induction, and the Universal Intelligence framework—center on the principle of simplicity: seeking the shortest possible program to reproduce a given data instance. This principle is formalized as **Kolmogorov Complexity (KC)**. As presented earlier, despite its intractability in theory, KC models an interesting signal for representation learning, enabling simpler / compressed representations. In this section, we draw analogies between adaptive tokenization & KC.

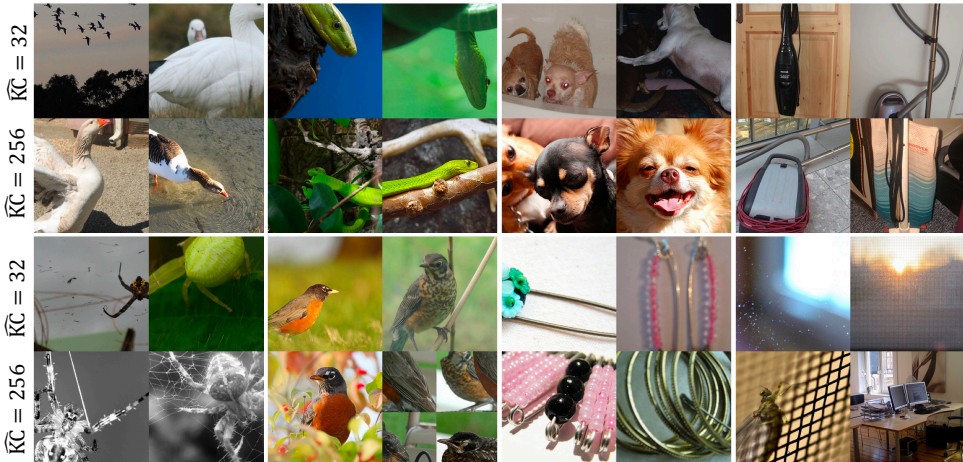

Figure 7: **Imagenet (IN) from Lens of Kolmogorov Complexity:** We bucketed images from different IN synsets based on their predicted approximate complexity $\hat{KC}$, with images sampled using $\hat{KC}_{\epsilon=0.07}(x, T = 256) = 32$ (row 1) or 256 (row 2). Aligned w/ human observation, KARL allocates fewer tokens to simpler images (row 1). **$\hat{KC}$ of Imagenet $\approx$ 146 tokens/image on average** (Tab. 2).

**Correlation with *intractable* Kolmogorov Complexity principle:** Kolmogorov Complexity (KC) formalizes the length of the *shortest program* that can *reproduce a given data instance*. In the context of adaptive tokenization, we approximate this notion by treating the number of tokens as a proxy for program length—each token representing a discrete, structured unit of information. Reducing the token count while still representing the input is thus analogous to minimizing its description length and hence its Kolmogorov Complexity. Let's understand this analogy in more detail.

More formally, a "**program**" for reproducing an image should include both the token sequence and the decoder that maps tokens back to the image. The encoder, in this view, acts more like a program synthesizer—searching for the token string that satisfies a reconstruction objective. Since the decoder is amortized across the dataset and shared across all samples, its contribution to the overall description length can be considered constant, and is therefore ignored in our analysis (we ablate the effect of varying decoder complexity later via Fig. 12 and Fig. 13). Secondly, **reproducing data** *perfectly* is not feasible in most deep learning settings. If we were to insist on exact reconstruction, the required token sequence—the program—would grow unbounded in length. Hence, we redefine the goal of "data reproduction" as achieving an image reconstruction within a specified error threshold (e.g., $\ell_1$ loss or perceptual loss), making minimal program length both tractable and meaningful in practice. This formulation also parallels the use of the Kolmogorov Structure Function (KSF) [31] in lossy compression (see Example II.4), where $KSF(x, \alpha)$ denotes the logarithm of the size of the smallest typical set $S$ containing $x$ such that the Kolmogorov complexity of $S$ ($KC(S)$) is at most $\alpha$. Here, $\alpha$ acts as a complexity budget, and the goal is to find a minimal set $S$ that sufficiently "explains" $x$—an idea analogous to finding the shortest token program that reconstructs $x$ within a target loss.

**Searching the shortest program—** Algorithmic Probability frameworks like Solomonoff Induction (SI) aim to find the simplest (i.e., shortest) program that explains observed data by brute-force enumerating over all programs which, when decoded using a universal prefix Turing machine, represent the data (or its prefix). Shorter programs are assigned higher probability, making SI a formal realization of Occam's Razor, an inductive bias toward simplicity. In high-level alignment with SI, modern adaptive tokenizers [19] approximate this search by varying the token budget at test time, iteratively increasing token count until the reconstructed image reaches a desired fidelity threshold.

The proposed KARL framework approximates this shortest token subset search through a one-shot selection mechanism: it predicts which subset of the input tokens is sufficient to reconstruct the image within a defined loss threshold (e.g., $\ell_1$ loss or perceptual loss). By conditioning the encoding process on the reconstruction constraint itself, KARL avoids exhaustive sampling of multiple hypotheses and instead adapts the representation in a single forward pass. This allows KARL to approximate the behavior of Solomonoff-style inference without the computational burden of iterative decoding.

The reconstruction threshold or the definition of data reproduction can be altered based on the downstream task by fine-tuning or training KARL accordingly. Finally, it is important to emphasize that these minimal token representations are only *approximations* of the formal theories, as deep

| Train Set | Approach | L1 Loss × 10 ↓ | | | LPIPS ↓ | | | SSIM ↑ | | | Dreamsim ↓ | | |
|---|---|---|---|---|---|---|---|---|---|---|---|---|---|
| | | 32 | 128 | 256 | 32 | 128 | 256 | 32 | 128 | 256 | 32 | 128 | 256 |
| IN100 | ALIT | 1.17 | 0.87 | 0.76 | 0.25 | 0.20 | 0.16 | 0.32 | 0.40 | 0.44 | **0.26** | **0.14** | **0.10** |
| | KARL | **1.13** | **0.80** | **0.70** | 0.30 | **0.19** | **0.14** | **0.34** | **0.44** | **0.49** | 0.28 | 0.16 | 0.11 |
| IN1K | One-D-Piece | 1.34 | 1.02 | 0.94 | 0.32 | 0.21 | 0.19 | 0.32 | 0.38 | 0.40 | 0.29 | 0.14 | 0.12 |
| | ALIT | 1.13 | 0.85 | 0.75 | 0.15 | 0.18 | 0.29 | 0.33 | 0.42 | 0.45 | **0.24** | **0.13** | **0.10** |
| | KARL | **1.10** | **0.81** | **0.68** | 0.29 | 0.18 | **0.14** | **0.35** | **0.44** | **0.50** | 0.27 | 0.14 | 0.11 |

Table 1: **Reconstruction Metrics on ImageNet100 using same token count for all images:** KARL performs equally well as ALIT and One-D-Piece, while being one-pass adaptive tokenizer.

| Approach | Tokens Used ↓ | | | Number of Enc + Dec Runs ↓ | | | LPIPS ↓ | | | SSIM ↑ | | |
|---|---|---|---|---|---|---|---|---|---|---|---|---|
| | 0.03 | 0.05 | 0.09 | 0.03 | 0.05 | 0.09 | 0.03 | 0.05 | 0.09 | 0.03 | 0.05 | 0.09 |
| ALIT | 251.5 | 227.2 | 136.6 | 7.86×2 | 7.01× 2 | 4.21× 2 | 0.16 | 0.17 | 0.22 | 0.44 | 0.43 | 0.39 |
| KARL | 252.9 | 229.6 | 152.5 | **1 + 1** | **1 + 1** | **1 + 1** | **0.14** | **0.15** | **0.21** | **0.49** | **0.47** | **0.42** |
| One-D-Piece | 255.1 | 247.9 | 178.4 | 7.97 + 1 | 7.74 + 1 | 5.57 + 1 | 0.19 | 0.19 | 0.22 | 0.40 | 0.40 | 0.38 |
| ALIT | 250.0 | 221.3 | 128.8 | 7.81 × 2 | 6.91 × 2 | 4.02 × 2 | 0.15 | 0.16 | 0.21 | 0.45 | 0.44 | 0.40 |
| KARL | 252.1 | 225.2 | 146.2 | **1 + 1** | **1 + 1** | **1 + 1** | **0.14** | **0.16** | **0.21** | **0.50** | **0.49** | **0.42** |

Table 2: **Recon. Metrics on IN100 sampling variable token count for different images:** All approaches perform equally well. Compared to ALIT & One-D-Piece, KARL can one-pass generate images satisfying a recon. loss threshold. First two rows are trained on IN100 vs rest on IN1K.

learning–based tokenizers provide *no guarantees of true minimality*. For instance, consider the Mandelbrot fractal shown in Fig. 5, which theoretically has extremely low Kolmogorov Complexity, yet still requires approximately 98 tokens to achieve a reconstruction loss around 0.05.

**Factors influencing learned estimates of minimum token count or Kolmogorov Complexity:** The proposed measure of image complexity depends on how well the tokenizer maps an image to its learned codebook space. Consequently, the estimates are influenced by whether the input is *in-distribution (IID)* or *out-of-distribution (OOD)*. For example, in Fig. 5, the dog and chessboard images illustrate this difference: despite the chessboard having a lower true KC, a tokenizer trained on ImageNet compresses dog image more rigorously due to its higher similarity to the training data. *Although this is an advantageous property of adaptive tokenizers—distinguishing IID vs OOD data* by assigning fewer tokens to IID, the gap with true KC can be reduced by training on larger datasets.

Another key factor is the amount of *structure versus noise* in the image. Like theoretical KC, our learned complexity measure is sensitive to noise—Kolmogorov Complexity does not distinguish between a highly structured image and pure noise; both can exhibit high complexity. Although higher-order AIT concepts such as *Sophistication* and *Logical Depth* aim to make this distinction, a detailed exploration of them is beyond our current scope. A preliminary way to separate structure from noise using adaptive tokenizers could be to analyze how image reconstruction metrics (e.g., SSIM, $\ell_1$) change with the number of tokens. For example, while KARL assigns the maximum token count (256) to both a chessboard and a noise image, SSIM improves significantly with token count for the chessboard due to its simple, redundant, and identifiable patterns—whereas SSIM remains largely unchanged for pure noise (Fig. 5). Aligned with the theoretical concepts of sophistication and logistic depth, more interesting images are those that exhibit a mix of predictable and unpredictable structure, often resulting in intermediate values of $\Delta$ reconstruction with increasing tokens. For instance, images like dogs or Mandelbrot sets (Fig. 5 rows 3,4) are more interesting to KARL than a simple chessboard or pure noise (Fig. 5 rows 1,2). This suggests that the rate of improvement in reconstruction quality as the token budget increases can serve as a heuristic for identifying meaningful structure, further reinforcing the potential alignment between adaptive tokenizers and AIT. A deeper investigation into these higher-order AIT concepts represents a promising direction for future research.

## 5 Experiments & Ablations

**Image Reconstruction:** We compare KARL against recent adaptive tokenization methods: **ALIT**—a recurrent tokenizer [19], **One-D-Piece**—a matryoshka-style method. These baselines are representative of other approaches like ElasticTok [21] and FlexTok [17], and enable a fair comparison since all three use a VQGAN-based 2D tokenizer, similar encoder/decoder architectures, token quantization, and GAN-based optimization—allowing us to focus the study specifically on minimum program search. See Fig. 1 for a conceptual comparison of KARL against these approaches.

We perform two types of comparison through Tab. 1 and Tab. 2. Tab. 2 reports reconstruction metrics across varying token counts, using the *same token count for all images*. All methods perform comparably, with KARL slightly outperforming on the majority of single-image metrics—L1, LPIPS,

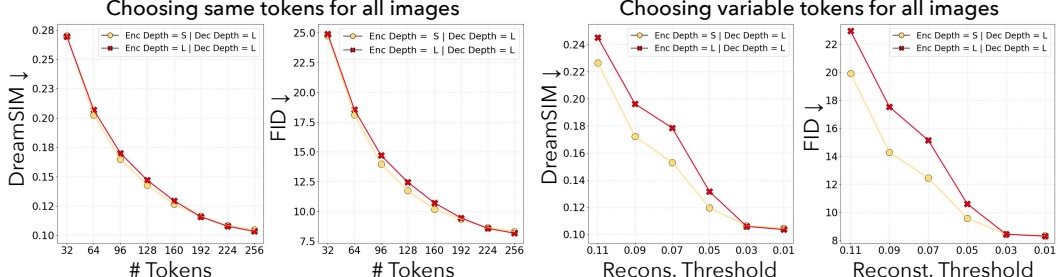

Figure 8: **Variable-token count allocation enables better scaling laws:** Mapping a highly complex image to a fixed, small token count is sub-optimal. The plots on the left use a fixed token count for all images, potentially suffering from such sub-optimality, leading to such poor scaling behavior compared to variable per-image allocation. **Scaling Law:** *Small Enc | Large Dec performs the best.*

and SSIM—while ALIT leads marginally on DreamSim. We report FID in the supplementary, as it poorly reflects per-image reconstruction and is misaligned with our goal of minimal program search under Kolmogorov Complexity (which seeks programs that represent data, not distributions). As shown in Fig. 4, visual comparisons highlight the strong reconstruction quality of KARL and ALIT over One-D-Piece, despite One-D-Piece slightly outperforming on FID.

In the second set of experiments shown in Tab. 2, we allocate a variable number of tokens per image based on a target L1 reconstruction loss condition. For ALIT, this requires multiple encoder-decoder runs until the reconstruction target is met — about $4\times$ more runs than KARL for a $0.09$ threshold, and up to $8\times$ more for lower thresholds. One-D-Piece, like other matryoshka-style methods, produces encodings in a single pass, but identifying the suitable representation still requires rolling out the decoder multiple times ($2$–$4\times$ on average over 5K IN100 images). In contrast, KARL produces a token embedding in a single pass that satisfies the reconstruction loss constraint, with only minor deviations from the target in a few cases. Metric-wise performance remains comparable across methods, but KARL offers the most practical one-pass inference. We believe its training strategy can be adapted for downstream and intelligent tasks, such as vision-language models (VLMs).

**Scaling Laws for Kolmogorov-Approximating Representation Learning:** The minimum program search for an image depends not only on the latent embedding but also on the decoder, which acts like a programming language. According to the invariance theorem in AIT [15], the true Kolmogorov Complexity (KC) includes a constant term tied to the decoder's design. The encoder, while not directly affecting KC, serves as the program synthesizer—making its capacity worth studying alongside the decoder. That said, since the encoder, decoder, and quantization codebooks are shared (amortized) across all images, we approximate image complexity using only the latent token count.

While prior work emphasizes the importance of decoder size [17, 32], we instead investigate the role of encoder capacity by comparing small vs. large encoder models (Fig. 8), keeping the decoder fixed. Please refer to the appendix for other scaling law ablations (Fig. 10, Fig. 11 and Fig. 12, Fig. 13). The left pair of plots allocates the same token count to all images and shows negligible difference between models. In contrast, the right pair allocates variable token counts based on a reconstruction loss threshold, revealing a clear advantage for the smaller encoder. The smaller encoder's superior performance is intuitive — the task of latent-distillation encoder is to simply distill VQGAN tokens to 1D tokens, a task that does not require high capacity. *But what enabled the right-pair experiments to surface this difference which the same per-image token allocation strategy failed to identify?* We attribute this to the variable token allocation strategy — letting the network choose the optimal token count itself—rather than evaluating every image under all token counts. It may even be suboptimal to test a highly complex image w/ fewer tokens. *Variable token allocation avoids such sub-optimal tests.* **Please refer to the Appendix for a lot more experiments.**

## 6 Conclusion

We introduce KARL – a one-pass adaptive image tokenization approach that dynamically allocates a variable fraction of the input token budget based on the complexity of an image. KARL introduces a novel training strategy, resembling upside-down reinforcement learning: it first attempts lossless compression, then uses the reconstruction delta as a guiding signal to learn when to halt the allocation of additional tokens. Our method achieves performance comparable to recent adaptive tokenization techniques across standard reconstruction metrics. Beyond empirical results, we offer a conceptual bridge between adaptive image tokenization and the seminal framework of Algorithmic Information Theory (AIT)—interpreting active token count as an approximation of Kolmogorov Complexity.

# 7 Acknowledgments

This work was supported by the Defence Science and Technology Agency, Singapore; Navy - ONR award N00014-22-1-2740, the Department of the Air Force Artificial Intelligence Accelerator and was accomplished under Cooperative Agreement Number FA8750-19-2-1000; and in part by the NSF Award 2019786 (The NSF AI Institute for Artificial Intelligence and Fundamental Interactions). We are grateful to LG Electronics for a funding gift and to Jung Guack (LG Electronics) for assisting Sanghbyun in running KARL on their cluster. We also thank our lab members — Chris Hill, Elliott, Erin (TIG), Ishaan, Jyo, Haoyu, Manel, Shamit, Tianwei, Tianyuan, Xingjian, and Yichen.

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

# A  Appendix

## A.1  Summarizing the Core Contributions

- **Single-pass adaptive tokenizer** that allocates variable-length representations per image.

  Why do we need an adaptive tokenizer that operates in a single pass? Modern large-scale models increasingly use hierarchical or token-based representations that offer flexibility in deployment, latency, or retrieval cost. Yet, in practice, adaptivity is often reduced to selecting a fixed number of tokens across all data, based solely on downstream task requirements. This is inefficient: complex inputs may be underrepresented, and simple ones overrepresented.

  We propose an adaptive tokenizer that predicts the number of tokens per image in a single forward pass, guided not only by task constraints but also by the image's intrinsic complexity. Our formulation enforces a target reconstruction quality (e.g., $\ell_1$ loss below a threshold), ensuring compact yet sufficient representations. Future extensions may incorporate perceptual similarity metrics or downstream-task feedback to further improve semantic adaptivity.

- **Loss-conditioned training strategy** inspired by KC and upside-down reinforcement learning.

  Prior adaptive methods [16, 19] rely on iterative inference-time procedures—searching over nested embeddings or using recurrent models—to identify the minimal token or memory budget required for a task. These approaches introduce latency and computational overhead.

  In contrast, we propose a loss-conditioned training strategy that enables the model to predict the appropriate number of tokens in a single forward pass. Drawing inspiration from Kolmogorov Complexity and the upside-down reinforcement learning paradigm [22], we treat reconstruction loss as a task-specifying condition. If an image can be reconstructed with $T$ tokens up to a given loss threshold, then using a larger budget $(T + \Delta T)$ should not be necessary. Our training procedure enforces this principle by conditioning the model on the loss obtained with $T$ tokens and optimizing it to match that quality using $T + \Delta T$ tokens—while masking out the redundant ones. This eliminates the need for test-time search and allows efficient, adaptive inference.

- Conceptual alignment of adaptive representations w/ **Algorithmic Information Theory (AIT)**.

  Our third contribution is a high-level conceptual bridge between hierarchical or adaptive tokenization and principles from AIT. We draw an intuitive correspondence: training models to halt once sufficient representational capacity is reached parallels the core idea of Kolmogorov Complexity. Moreover, the progressive allocation of tokens in hierarchical tokenizers echoes second-order AIT notions such as *sophistication* and *logical depth*. We stress that this is a conceptual perspective—not a formal theoretical result—and offer it as a starting point for further exploration of the intersection between AIT and adaptive representation learning.

## A.2  Experimental Details

**Datasets:**   Throughout the paper, we utilize Imagenet or a 100-class subset of Imagenet dataset to train and evaluate our models. The 100-class subset of Imagenet is utilized in multiple prior works, serving a good dataset with decent scale (0.1M images) and lesser compute requirements. Human evaluation (Fig. 14) was done on Savioas Dataset [33] which contains human annotations of complexity on several known computer vision datasets.

**Compute Requirements:**   Majority of the training was done on single A100 or H100 machine with eight 80GB gpus or on machines with equivalent GPU memory (distributed training on 4 H200 gpus).

## A.3  Training Procedure

Our training framework follows a two-phase formulation as described in the main paper: **(1) Estimating Image Complexity**, where the model attempts near-lossless reconstruction using a randomly-sampled token budget $T$, and **(2) Learning to Tokenize at Estimated Complexity**, where the model learns to ignore surplus tokens when provided a larger token budget $T + \Delta T$, while maintaining the same reconstruction quality. These two phases are jointly optimized with complementary objectives:

- *Lossless Compression Objective*: Encourages the model to fully utilize the given token budget to minimize reconstruction and quantization losses.

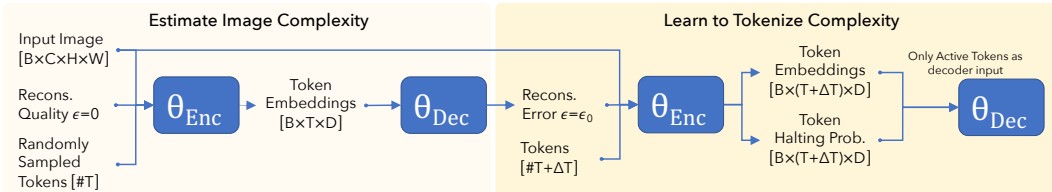

Figure 9: **KARL Training Graph:** KARL follows an upside-down RL paradigm—(a) **Estimate Image Complexity** stage samples task-defining inputs {image, token budget T, reconstruction error $\epsilon_0$} by attempting lossless compression ($\epsilon = 0$). (b) **Learn to Tokenize** stage trains the model—conditioned on $\epsilon_0$—to match the same quality using T+$\Delta$T tokens while halting the extra (rightmost) $\Delta$T. The shared encoder $\theta_{\text{Enc}}$ and decoder $\theta_{\text{Dec}}$ are updated in each iteration using reconstruction losses in both stages and halting loss in the second.

- *Minimal Program Search Objective*: Encourages the model to identify and suppress unnecessary tokens via halting supervision, enabling it to approximate the shortest sufficient representation.

Together, these objectives guide the model to produce variable-length token representations that reflect the intrinsic complexity of each image, enabling one-shot adaptive tokenization without iterative search at inference. See Fig. 9, Fig. 3 for the training computational graph and the training algorithm.

**Loss-Conditioned Training Details:** Our training procedure reflects the upside-down reinforcement learning (RL) paradigm: reconstruction loss conditions act as task-specifying rewards that guide the learning process. We initialize a list of discrete $\ell_1$ loss targets—including low values (0.0, 0.01, 0.02), mid-range values (0.03–0.11), and a few larger ones (0.14–0.4)—each mapped to an embedding vector of the same dimensionality as the encoder's input tokens. These loss embeddings are concatenated as an additional token to the latent-distillation encoder input.

In each training iteration, a token count is randomly sampled from 16, 32, ..., 256. In the first run, the model attempts near-lossless compression using a small loss condition and token budget $T$, producing an actual reconstruction error $\epsilon_0$. We then map $\epsilon_0$ to the smallest greater discrete loss from the list and use the corresponding embedding to condition the second run, which uses a larger token budget ($T + \Delta T$, up to 256).

The token halting probabilities are supervised using a binary cross-entropy loss: tokens from the first run are labeled "keep" (0) and new tokens are labeled "halt" (1). If no tokens are added (i.e., $T = 256, \Delta T = 0$), we augment training by randomly sampling loss values that are smaller than the one attained during the lossless run. This ensures the model still receives diverse training pairs of (target loss condition, input token budget), even when no masking occurs in the second stage. This formulation allows the model to learn how to tokenize an image given a task-specific loss condition, effectively converting a token allocation problem into a supervised learning task—analogous to how upside-down RL uses rewards as inputs to learn policies.

Tab. 3 evaluates the learned tokenizer's ability to meet the input reconstruction loss condition—specifically, how often the reconstructed image satisfies the input loss condition when a portion of the input tokens is masked. When no tokens are masked, some images may still exceed the loss threshold due to an insufficient max token budget (=256). As shown in the table, only a very small fraction of the 5000 IN100 validation images exhibit a reconstruction loss greater than 0.03 beyond the input threshold when masking is applied. For images that utilize only a subset of the 256 input tokens yet exceed the input loss condition ($\epsilon$), the average reconstruction error is only marginally above the threshold. These results validate the efficacy of our training approach.

**Details regarding multi-stage training pipeline:** Inspired by recent works [26, 19], our overall training is conducted in two *distinct stages over the course of training epochs*, separate from the two-phase EIC–LTC procedure executed within each training iteration. Specifically, this multi-stage pipeline consists of a **latent-distillation pretraining** followed by a **GAN-based finetuning stage**.

In the latent distillation stage, a pre-trained image tokenizer (VQGAN or VAE) is used to map input images into 2D token grids. The KARL encoder and decoder modules are trained to compress these 2D tokens into 1D representations and reconstruct them back, using reconstruction loss over the 2D token space as the primary learning signal (thus performing 2D latent tokens $\rightarrow$ 1D distillation).

| Threshold | % of images with recon. loss different from input condition ($\epsilon$) | | | | | | Avg Err ($> \epsilon$ only) |
|---|---|---|---|---|---|---|---|
| ($\epsilon$) | $> \epsilon$ | $> \epsilon+0.01$ | $> \epsilon+0.02$ | $> \epsilon+0.03$ | $> \epsilon+0.04$ | $> \epsilon+0.05$ | |
| 0.01 | 0% | 0% | 0% | 0% | 0% | 0% | 0.000 |
| 0.03 | 1% | 0% | 0% | 0% | 0% | 0% | 0.050 |
| 0.05 | 10% | 3% | 1% | 1% | 0% | 0% | 0.060 |
| 0.07 | 25% | 12% | 6% | 4% | 2% | 1% | 0.086 |
| 0.09 | 18% | 9% | 5% | 3% | 2% | 1% | 0.107 |
| 0.11 | 36% | 19% | 10% | 5% | 3% | 2% | 0.126 |

Table 3: **Analysis of the loss-conditioned training strategy for single-pass adaptive tokenization:** Only a small fraction of images exceed the input target reconstruction loss threshold ($\epsilon$) **when masking is applied**. The columns indicate the fraction of images exceeding $\epsilon$ by more than specified margin values. The last column shows the average reconstruction error for these images, which remains only marginally higher than the input threshold. Default $\epsilon = 0.05$ works well at inference.

Both encoder and decoder operate with full self-attention: the encoder attends over the concatenated sequence of image tokens and initialized 1D tokens, while the decoder attends over learned 1D tokens concatenated with masked 2D tokens (in an inpainting-style setup). For VQGAN tokenizers, we use a cross-entropy loss over predicted logits and ground-truth codebook indices. For VAE tokenizers, we use mean squared error (MSE) loss over the predicted and target token embeddings.

In the second stage, we finetune the model using pixel-level losses: reconstruction loss shifts from the intermediate 2D token space to the final image space. This includes pixel-wise $\ell_1$ loss, an adversarial GAN loss, LPIPS perceptual loss, and standard quantization losses (commitment and codebook loss). Following prior work, we apply quantization on a factorized 12-dimensional embedding derived from the encoder output. Finally, given recent success of diffusion-based autoencoders instead of GANs [27, 17], extending KARL w/ diffusion-based objectives will be promising as future work.

## A.4 Scaling Laws & Ablations

Most adaptive tokenization works [19, 17] present scaling laws by varying the number of tokens per image, but they typically assign the same token count to all images for a given configuration. As noted in Core Contribution #1, the ideal goal of an adaptive tokenizer is to allocate a variable number of tokens per image at test time, depending on the content or complexity of each image. To capture both perspectives, we design scaling laws in two ways: (a) uniform token allocation across images, as done in prior work, and (b) variable token allocation based on a target reconstruction loss. Additionally, inspired by the Algorithmic Information Theory (AIT) notion of Kolmogorov Complexity, we analyze scaling laws from a KC perspective—studying how well we can minimize the token count (i.e., program length) per image while preserving recons. quality, target $\ell_1$ loss.

Before delving into the scaling laws, we clarify an important detail: ideally, the Kolmogorov Complexity (KC) of a data point includes the full length of the program required to generate it. In our context, this would encompass not only the number of encoded tokens but also the token feature dimensionality, decoder size, and codebook size. However, since these latter components are amortized across many images, we approximate KC using only the token count and refer to this as $\hat{\text{KC}}$ throughout the paper. Note that the encoder is not part of the generative program—it acts as a program synthesizer rather than the program itself.

**Continuous vs Discrete Tokenization:** We compare continuous and discrete adaptive tokenization approaches at both 1D and 2D token levels under both fixed and variable token allocation regimes. Key observations from Fig. 10 include  (new or significant findings highlighted in green) :

- Continuous 1D tokens consistently outperform quantized 1D tokens in terms of reconstruction quality (measured by FID) across nearly all token counts—echoing observations of prior works.

- This performance advantage holds even when token counts are allocated adaptively per image based on a target reconstruction loss.

- Continuous 1D tokenizers achieve better reconstruction at lower average token counts (i.e., lower estimated $\hat{\text{KC}}$), highlighting their efficiency in capturing image complexity.

- The choice of 2D tokenizer (continuous VAE vs. discrete VQGAN has *minimal* impact on the minimum token count required to reach a target loss—*when the 1D tokens are quantized.*

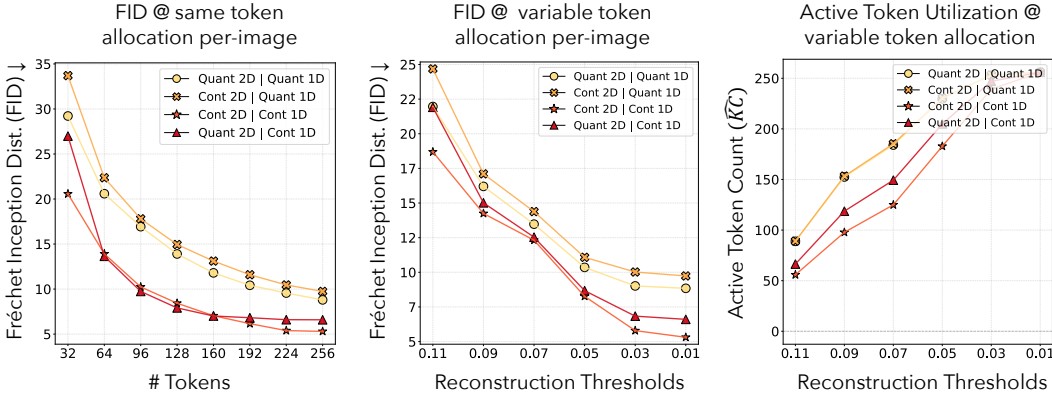

Figure 10: **Scaling laws for continuous vs discrete tokenization:** Cont1D is most crucial for FID (Plot 1 & 2). Cont2D (VAE) with Cont1D leads to shortest representation ($\hat{KC}$) on average (Plot 3).

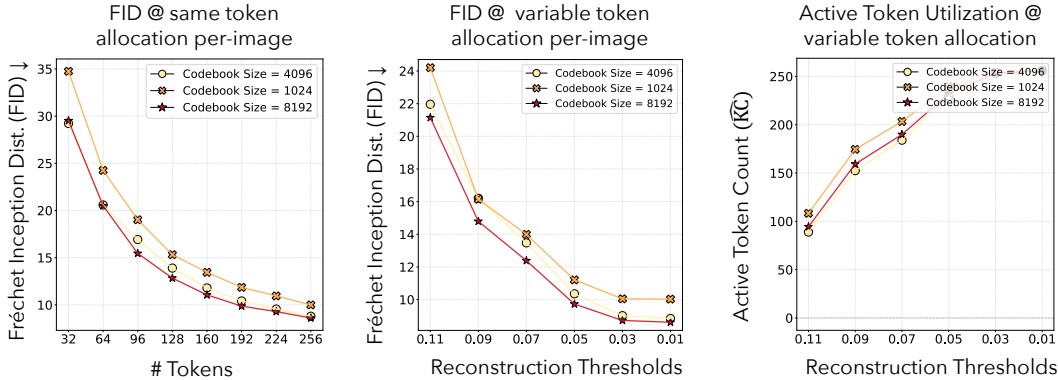

Figure 11: **Scaling laws for 1D token codebook size:** Larger codebooks (4096 and 8192 codes) lead to shorter representation length ($\hat{KC}$) on average because of code specialization.

- However, when 1D tokens remain continuous, pairing them with a discrete 2D tokenizer (e.g., VQGAN) leads to *higher* $\hat{KC}$ across all target loss levels, potentially because more tokens are required to compensate for the loss of detail at 2D token level.

**1D Token Codebook Size Ablations:** From Fig. 11, we observe that increasing the codebook size consistently improves performance, as reflected by lower (i.e., better) FID scores. Moreover, larger codebooks (4096 and 8192) not only achieve better image quality, but do so using fewer tokens on average—resulting in lower estimated Kolmogorov Complexity ($\hat{KC}$) per image. This is intuitive as smaller codebooks offer fewer specialized codes, requiring more tokens to achieve the same representational fidelity.

**Encoder and Decoder Depth / Width Ablations:** As highlighted in the main paper and shown in Fig.12 and Fig.13, the scaling laws indicate that the best-performing architecture features a *small encoder and a large decoder*. This is intuitive: the encoder's role is to distill the already abstracted 2D tokens from the VQGAN into 1D tokens, while the decoder takes on the more challenging task of inpainting 1D tokens into a masked slate.

A key observation is that the variable token allocation strategy enhances the separation between architectural variants. Specifically, it reveals a more pronounced performance gap between the Small Encoder, Large Decoder and Large Encoder, Large Decoder configurations. When the same number of tokens is assigned to all images — regardless of image complexity — high-complexity images may perform poorly under limited token budgets, making it harder to distinguish between architectural strengths. In contrast, when token counts are adaptively allocated per image, each model can adjust token usage to meet reconstruction targets, enabling fairer comparison and clearer differentiation.

The average number of active tokens remains nearly constant across ablations. The Small Encoder, Large Decoder setup (small in both width and depth) performs best under this similar active token

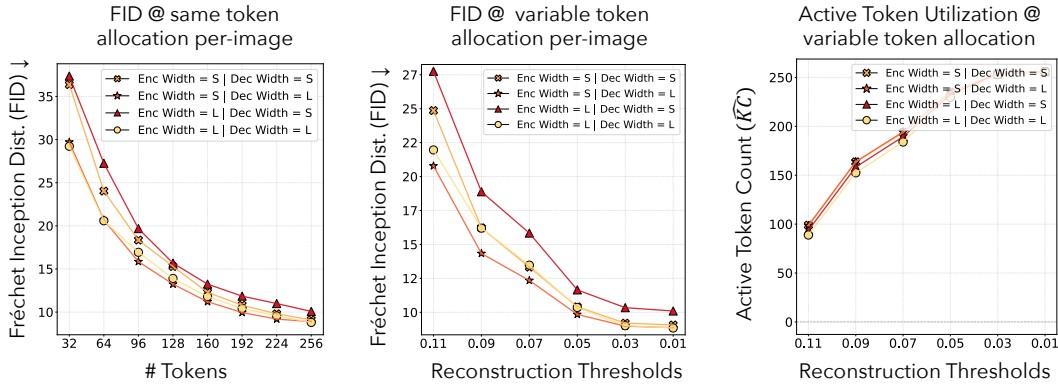

Figure 12: **Scaling laws for encoder/decoder width (feature dimensionality):** A *Small Encoder with a Large Decoder* consistently achieves the best performance. Notably, variable token allocation per image enables a more informative comparison, clearly distinguishing the *Small Encoder, Large Decoder* configuration from the *Large Encoder, Large Decoder* variant.

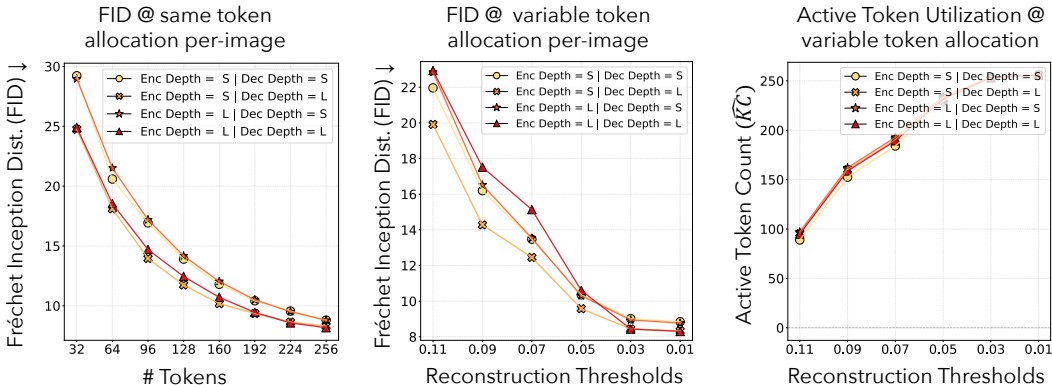

Figure 13: **Scaling laws for encoder/decoder depth (number of layers):** Similar to the width ablation, the *Small Encoder with a Large Decoder* configuration yields the best performance at the same average active token count as other variants. This distinction becomes more evident when using variable token allocation per image, as opposed to fixed token counts.

utilization. **Note:** For the width ablations, both the depth (i.e., number of transformer layers) and the number of heads (in multi-head attention) are held constant. Conversely, in the depth ablations, the width is fixed to a large value, and the number of heads scales proportionally with the depth.

### A.5  Alignment with human measures of complexity

Fig. 14 showcase strong alignment between human labels of complexity and our predicted estimate – min. token count per image or approx $\hat{KC}$. We computed $\hat{KC}$ for images from the Savoias dataset [33] using $\epsilon = 0.05$ as input reconstruction condition – $\hat{KC}_{\epsilon=0.05}(x, T = 256)$, followed by bucketing them into intervals of 32. Around $47\%$ of the images with complexity labeled 15 (/ 100) by humans are assigned 32 to 64 tokens, while images with human-assigned complexity measures of 60+ get $\hat{KC} \approx 256$.

### A.6  Additional Comparison with Prior Adaptive Tokenizers

Beyond the comparisons presented in the main paper, we further evaluate KARL against ALIT, a recurrent adaptive tokenizer, and One-D-Piece, a Matryoshka-based tokenizer, under both fixed per-image token allocation and variable token allocation regimes.

Fig. 15 and Fig. 16 clearly demonstrate the superior performance of ALIT and KARL over One-D-Piece, which frequently produces blurry reconstructions (see the second column in both figures). Such blurry outputs misleadingly yield good FID scores (see Tab. 6) while performing poorly on per-image metrics such as LPIPS, SSIM, PSNR, and DreamSIM. This observation *underscores a limitation of FID: it may not be an ideal metric when the focus is on accurate per-image reconstruction. Furthermore, from a Kolmogorov Complexity (KC) perspective, the goal of tokenization is to*

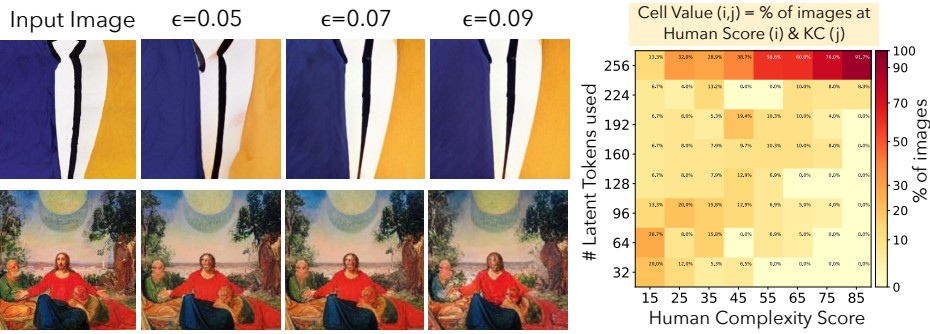

Figure 14: **Alignment of learned image complexity with human estimates on Savoias [33]**: The top row is marked as less complex (8) by human compared to bottom row (89). Active token counts or K̂C (Row 1 = 64,32,32 | Row 2 = 256,256,192). Heatmap shows large % of high complex images require large token counts, while a large % of low complex images utilize token counts from 32 to 64.

*produce the shortest representation that faithfully captures the individual data point—not merely the distribution.* In this regard, KARL achieves the best performance on most per-image reconstruction metrics, slightly outperforming ALIT. Unlike ALIT, which requires multiple recurrent encoder passes, KARL determines the minimal token count in a single forward pass.

Fig. 17 and Fig. 18 compare different adaptive tokenizers under a variable token allocation regime, where each image is assigned a different number of tokens based on a target reconstruction loss threshold. For One-D-Piece, this involves a single encoder pass but multiple decoder runs to find the shortest embedding that achieves reconstruction $\ell_1$ loss below the target. In contrast, ALIT performs a joint encoder-decoder pass at each iteration, requiring multiple iterations until the stopping criterion is satisfied. See Tab. 2 in the main paper for additional details.

Thanks to its recurrent training and explicit test-time search, ALIT performs best under a relaxed reconstruction criterion of $\ell_1$ loss $< 0.09$ (as shown in Fig. 17). However, KARL performs competitively—often surpassing ALIT on per-image metrics such as LPIPS, SSIM, and $\ell_1$ loss.

Fig. 18 analyses reconstructions under a tighter constraint of $\ell_1$ loss $< 0.05$. KARL not only achieves better or comparable performance on these metrics but also does so with equal or fewer tokens, particularly when compared to One-D-Piece. Based on our experiments, we conjecture that KARL with an input condition of $\epsilon = 0.05$ represents the most practical configuration for adaptive tokenization in real-world settings.

### A.7 Evaluating Downstream Tasks

We perform a range of downstream evaluation for CLIP text-image alignment (Tab. 8 and Tab. 9), depth estimation (Tab. 10 and Tab. 11), semantic segmentation (Tab. 12 and Tab. 13) and VLM (Tab. 14) tasks, by levering the reconstructions from different adaptive tokenizers. We evaluate both "fixed same-token for each dataset image" setting (Tab. 8, Tab. 10, Tab. 12, Tab. 14) and "variable token" setting based on desired reconstruction loss threshold $\epsilon$ (Tab. 9, Tab. 11, Tab. 13). In these tables, the 256-token, $\epsilon = 0.0$ column corresponds to the setting where the input token budget is 256 and meaning all tokens are used for lossless reconstruction.

Our findings reveal that **ALIT is the strongest adaptive tokenizer** across most metrics, using a recurrent transformer that incrementally adds tokens—each step incurring extra FLOPs. **KARL generally consistently ranks second, followed by the matryoshka-style One-D-Piece.** Notably, all three perform comparably, and KARL matches this performance in a single pass, highlighting its promise as a efficient, scalable alternative. A compelling future direction is to combine KARL with ALIT-style recurrent refinement—refining minimal representation with more iterative loops.

As shown across CLIP, Depth, and Semantic Segmentation metrics, the performance drop from $\epsilon = 0.0$ to $\epsilon = 0.03$ and $\epsilon = 0.05$ is minimal—highlighting the **benefit of KARL: allowing slight reconstruction loss has little to no impact on downstream performance.** As expected, the effect of increasing $\epsilon$ is especially marginal for coarse tasks like CLIP alignment.

**Note:** Compared to KARL and ALIT, One-D-Piece requires 30+ more tokens to achieve -bounded reconstructions with similar downstream performance (see Tab. 2 in the main paper).

Overall, the metrics consistently highlight the significance of KARL—achieving minimal or no drop in performance while operating in a single pass.

### A.8 Linear Probing on learned token representation

We conducted linear probing experiments by training a single-layer linear classifier on top of the latent tokens produced by KARL and ALIT. Using only the first 32 1D latent tokens, *ALIT achieves 44.8% Top-1 classification accuracy on ImageNet-1K, while KARL attains 41.5%*. When both the 2D image tokens and the first 32 1D tokens are used, performance improves to *51.6% for ALIT and 45.0% for KARL—surpassing with the non-adaptive Titok tokenizer (48.0%)*.

As noted in the ALIT paper, linear-probing accuracy improves with more recurrent iterations during training—potentially due to added training FLOPs and hierarchical gradient flow across RNN layers. This may partly explain ALIT's edge over KARL. Finally, investigating how compression or minimality impacts the linear separability of learned tokens is a promising direction for future work.

### A.9 Further Details on Computational Requirements

Tab. 2 compares computational complexity in terms of the number of encoder/decoder calls, which is architecture-agnostic and avoids confounds like GPU type or precision. For fair comparison, all baselines use the same encoder/decoder architecture. Below are GFlops and wall-clock runtimes observed on a single H100 GPU with FP32 precision.

**GFlops:**
- Encoder: KARL $\approx$ One-D-Piece $\approx$ First iteration of ALIT $\approx$ **80 GFlops.**
- Decoder: ALIT and One-D-Piece require multiple decoding passes, each $\sim$ **80 GFlops.**
- ALIT: Requires 8 iterations during training, and $\sim 4$ iterations at test time for -bounded reconstruction. Each additional ALIT iteration adds $\sim$ **30 GFlops**.

**Wall-clock Time (ms):** First encoder or decoder pass: $\sim$**7 ms**. Each additional ALIT encoder iteration: $\sim$**4 ms**.

**Figures and Tables in the following pages.**

| Method | L1 ↓ | | | | SSIM ↑ | | | | LPIPS ↓ | | | |
|---|---|---|---|---|---|---|---|---|---|---|---|---|
| | 32 | 64 | 128 | 256 | 32 | 64 | 128 | 256 | 32 | 64 | 128 | 256 |
| Flextok-vae-quant-1D | 1.75 | 1.51 | 1.20 | 0.98 | 0.23 | 0.27 | 0.33 | 0.40 | 0.53 | 0.48 | 0.42 | 0.36 |
| KARL-vae-quant-1D | **1.14** | **0.98** | **0.84** | **0.70** | **0.33** | **0.38** | **0.43** | **0.50** | **0.40** | **0.35** | **0.29** | **0.23** |

Table 4: **Comparison with Flextok:** KARL-S outperforms Flextok-S on L1, SSIM & LPIPS metrics.

| Eval Set | Approach | L1 × 10 ↓ | | | LPIPS ↓ | | | SSIM ↑ | | |
|---|---|---|---|---|---|---|---|---|---|---|
| | | 32 | 64 | 256 | 32 | 64 | 256 | 32 | 64 | 256 |
| COCO | One-D-Piece | 1.37 | 1.18 | 0.95 | 0.48 | 0.41 | 0.33 | 0.33 | 0.36 | 0.41 |
| | ALIT | 1.16 | 0.98 | 0.74 | 0.42 | 0.36 | 0.27 | 0.34 | 0.38 | 0.46 |
| | KARL | **1.11** | **0.96** | **0.64** | **0.41** | **0.35** | **0.23** | **0.36** | **0.40** | **0.53** |
| WIT | One-D-Piece | 1.13 | 0.96 | 0.75 | 0.31 | 0.24 | 0.16 | 0.45 | 0.47 | 0.52 |
| | ALIT | 0.92 | 0.77 | 0.57 | 0.37 | 0.31 | 0.24 | 0.46 | 0.51 | 0.58 |
| | KARL | **0.86** | **0.73** | **0.49** | **0.25** | **0.20** | **0.12** | **0.49** | **0.53** | **0.65** |

Table 5: **Reconstruction metrics on OOD datasets** (Top) Trained on IN1K (object-level dataset), evaluated on COCO (scene-level dataset). (Bottom) Trained on IN1K, evaluated on Wikipedia Image Dataset (more OOD). KARL demonstrates impressive generalization on OOD reconstructions.

| Approach | FID @ variable token regime (↓) | | | FID @ same token regime (↓) | | |
|---|---|---|---|---|---|---|
| | 0.03 | 0.05 | 0.09 | 32 | 128 | 256 |
| One-D-Piece | **8.2** | **8.3** | **10.5** | 24.1 | **9.7** | 8.2 |
| ALIT | 8.4 | 9.3 | 13.5 | **22.6** | 11.7 | **8.2** |
| KARL | 9.1 | 10.3 | 16.2 | 28.7 | 13.6 | 8.8 |

Table 6: **FID Comparison on IN100:** While KARL slightly trails prior approaches in FID, it outperforms them on single-image metrics that better align with the Kolmogorov Complexity (KC) perspective—favoring the shortest possible representation for each individual data point rather than alignment with dataset-level statistics. As is well known, FID is not a reliable metric for assessing reconstruction quality. Visual results in Fig. 4, Fig. 15, Fig. 16, Fig. 17, and Fig. 18 clearly demonstrate that both ALIT and KARL produce reconstructions that are visually superior to One-D-Piece. Notably, KARL is the only single-pass method for adaptive tokenization and achieves the best performance on single-image quality metrics. FID could likely be improved through additional training techniques—for instance, using diffusion decoders or separate (well-tuned) GAN discriminators for different representation lengths/token counts, instead of a shared one. Conditioning masking on perceptual losses like LPIPS, DreamSim (instead of $\ell_1$) might enhance FID.

| Method | FID ↓ | | | |
|---|---|---|---|---|
| | 32 | 64 | 128 | 256 |
| Flextok-vae-quant-1D | **25.28** | **21.95** | 19.58 | 17.26 |
| KARL-vae-quant-1D | 33.70 | 22.36 | **14.95** | **9.74** |

Table 7: **Comparison with Flextok on FID metric.**

| Approach | 32 tokens | 64 tokens | 256 tokens |
|---|---|---|---|
| One-D-Piece | 82.1 | 86.7 | 91.5 |
| ALIT | **84.6** | **88.5** | **92.8** |
| KARL | 82.3 | 86.7 | 91.2 |

Table 8: **Downstream Evaluation – CLIP text–image** (fixed token per-image).

| Approach | 256, $\epsilon = 0.00$ | 256, $\epsilon = 0.03$ | 256, $\epsilon = 0.05$ | 256, $\epsilon = 0.09$ |
|---|---|---|---|---|
| One-D-Piece | 91.5 | 91.5 | 89.8 | **89.6** |
| ALIT | **92.8** | **92.7** | **92.2** | **89.6** |
| KARL | 91.2 | 91.7 | 90.7 | 87.1 |

Table 9: **Downstream Evaluation – CLIP text–image** (input token budget: 256 tokens; subset of tokens selected such that reconstruction loss less than recon. threshold $\epsilon$). Higher is better.

| Approach | 32 tokens | 64 tokens | 256, $\epsilon = 0.0$ |
|---|---|---|---|
| One-D-Piece | 2.73 | 2.11 | 1.53 |
| ALIT | **2.04** | **1.62** | 1.17 |
| KARL | 2.19 | 1.74 | **1.16** |

Table 10: **Downstream Evaluation – Depth Estimation** (fixed token per-image).

| Approach | 256, $\epsilon = 0.0$ | 256, $\epsilon = 0.03$ | 256, $\epsilon = 0.05$ | 256, $\epsilon = 0.09$ |
|---|---|---|---|---|
| One-D-Piece | 1.53 | 1.53 | 1.55 | 1.77 |
| ALIT | 1.17 | **1.18** | **1.24** | **1.51** |
| KARL | **1.16** | **1.18** | 1.32 | 1.76 |

Table 11: **Downstream Evaluation – Depth Estimation** (input token budget: 256 tokens; subset of tokens selected such that recon. loss less than reconstruction threshold $\epsilon$. Lower is better.)

| Approach | F-measure @ 32 | F-measure @ 64 | F-measure @ 256 |
|---|---|---|---|
| One-D-Piece | 66.46 | 73.36 | 79.79 |
| ALIT | **70.58** | **75.96** | **81.55** |
| KARL | 68.09 | 73.18 | 80.64 |

Table 12: **Downstream Evaluation – Semantic Segmentation Estimation** (fixed token per-image).

| Approach | F-measure ($\epsilon = 0.03$) | F-measure ($\epsilon = 0.05$) | F-measure ($\epsilon = 0.09$) |
|---|---|---|---|
| One-D-Piece | 79.77 | 79.54 | **77.17** |
| ALIT | **80.90** | **79.95** | 76.27 |
| KARL | 80.49 | 78.88 | 73.41 |

Table 13: **Downstream Evaluation – Semantic Segmentation** (input token budget: 256 tokens; subset of tokens selected such that recon. loss less than reconstruction threshold $\epsilon$. Higher is better.)

| Approach | VLM Score (32 tokens) | VLM Score (64 tokens) | VLM Score (256 tokens) |
|---|---|---|---|
| One-D-Piece | 62.4 | 71.4 | 76.7 |
| ALIT | **64.1** | **71.6** | 77.0 |
| KARL | 63.8 | 68.4 | **77.6** |

Table 14: **VLM (VQA) downstream task.** Higher is better.

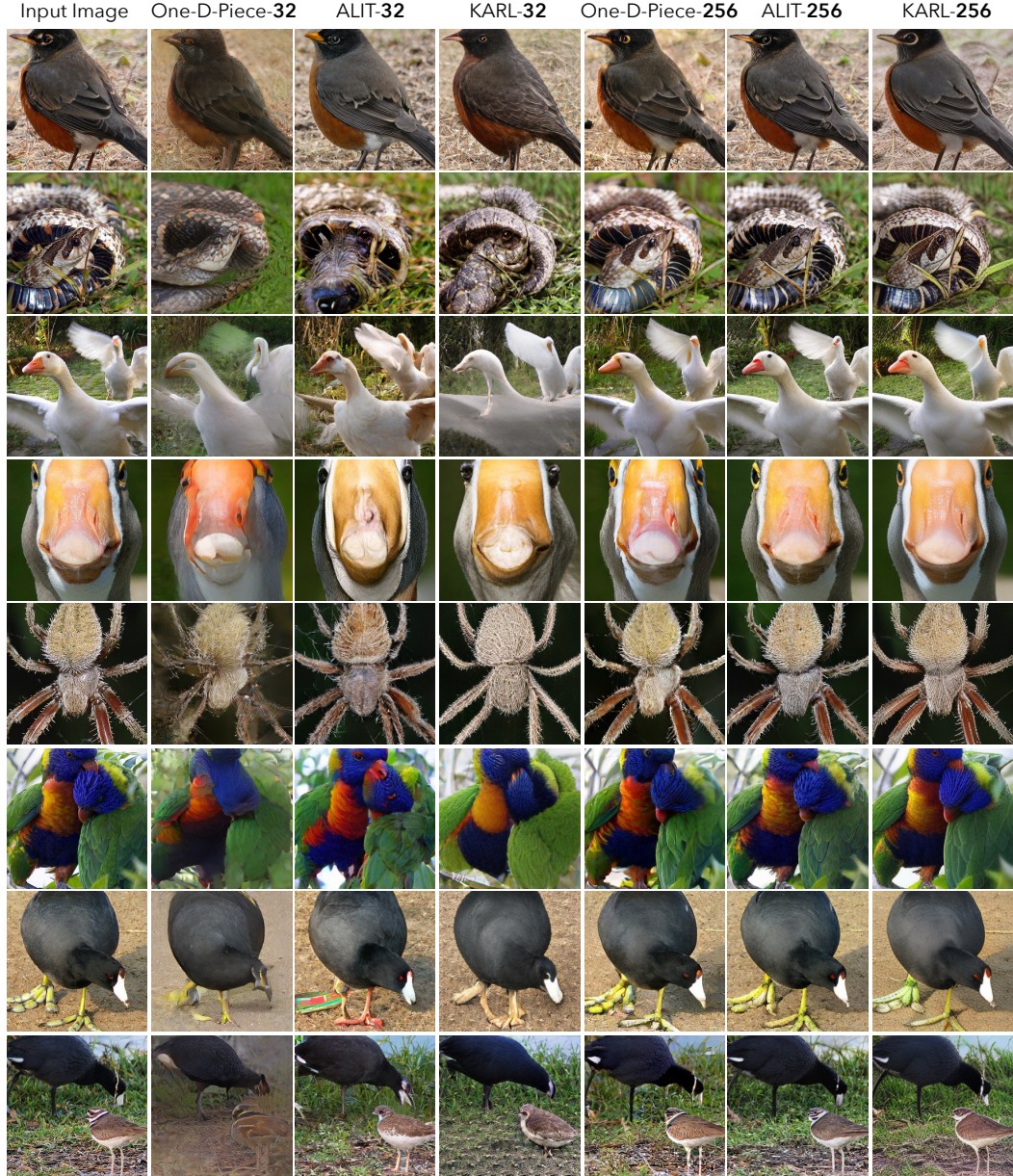

Figure 15: **Reconstruction Comparison on IN100 (visualization** $1/2$**):** KARL and ALIT produce higher-quality reconstructions than One-D-Piece, with KARL achieving strong single-image metrics. One-D-Piece consistently generates blurry images at low token counts satisfying FID metric more.

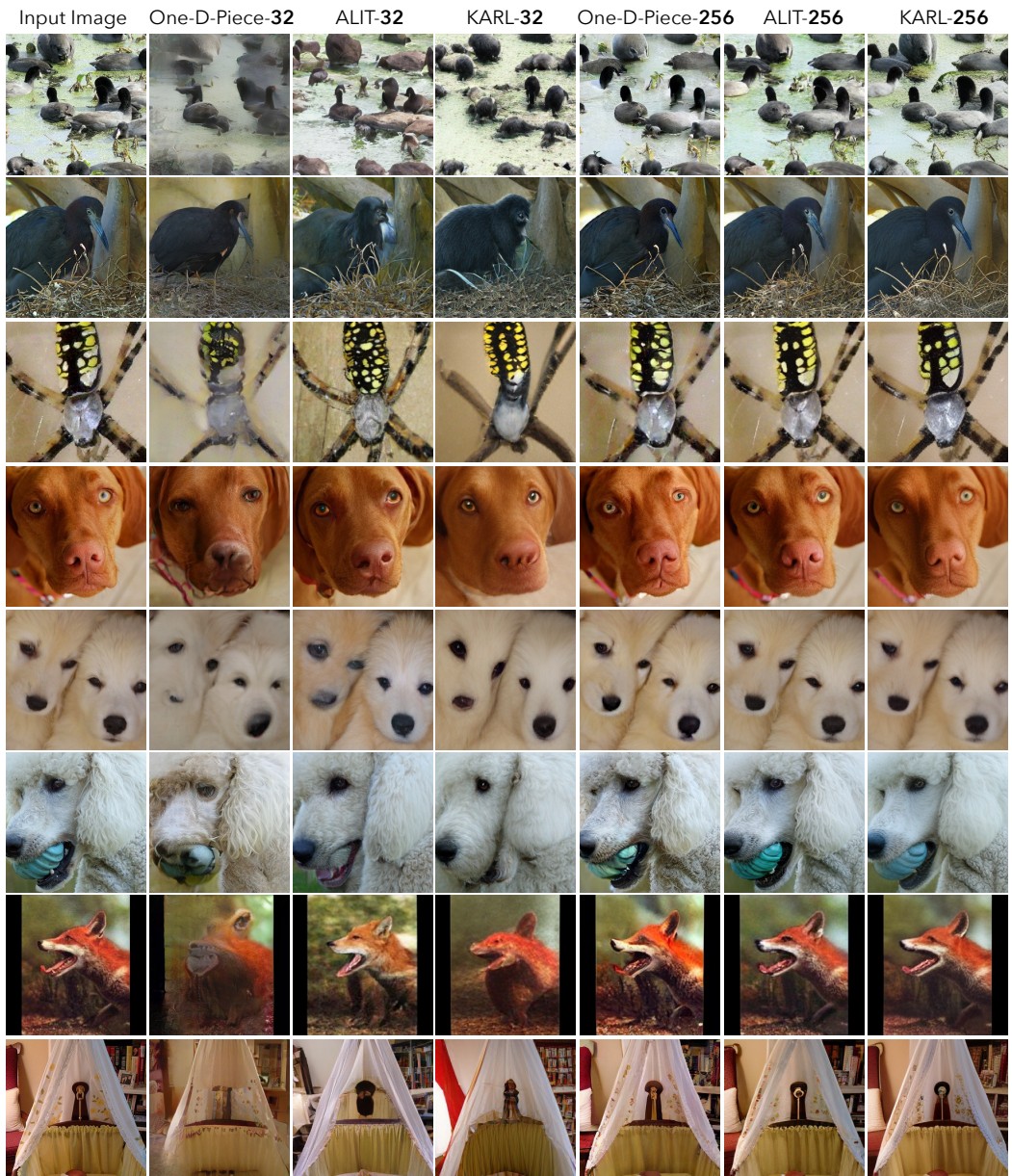

Figure 16: **Reconstruction Comparison on IN100 (visualization** 2/2**):** KARL and ALIT produce higher-quality reconstructions than One-D-Piece, with KARL achieving strong single-image metrics. One-D-Piece consistently generates blurry images at low token counts satisfying FID metric more.

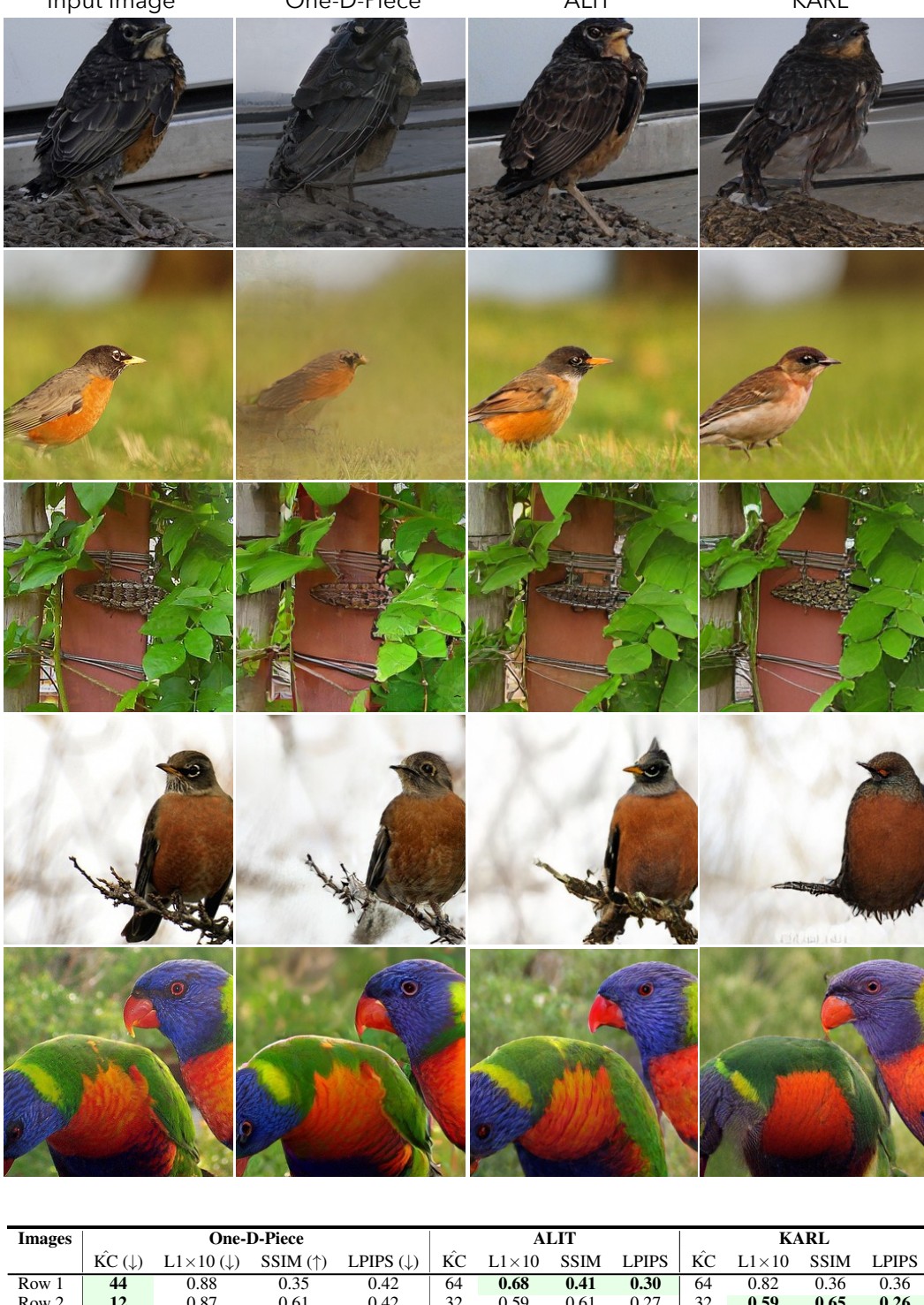

| Images | One-D-Piece | | | | ALIT | | | | KARL | | | |
|--------|------|------|------|------|------|------|------|------|------|------|------|------|
| | K̂C (↓) | L1×10 (↓) | SSIM (↑) | LPIPS (↓) | K̂C | L1×10 | SSIM | LPIPS | K̂C | L1×10 | SSIM | LPIPS |
| Row 1 | **44** | 0.88 | 0.35 | 0.42 | 64 | **0.68** | **0.41** | **0.30** | 64 | 0.82 | 0.36 | 0.36 |
| Row 2 | **12** | 0.87 | 0.61 | 0.42 | 32 | 0.59 | 0.61 | 0.27 | 32 | **0.59** | **0.65** | **0.26** |
| Row 3 | 256 | 1.07 | 0.27 | 0.40 | **192** | 0.89 | 0.31 | 0.36 | 224 | **0.83** | **0.37** | **0.30** |
| Row 4 | 75 | 0.90 | **0.59** | **0.32** | **32** | **0.85** | 0.58 | 0.35 | 64 | 1.00 | 0.59 | 0.37 |
| Row 5 | 256 | 0.92 | 0.32 | 0.34 | **128** | **0.79** | **0.32** | **0.34** | **128** | 0.87 | 0.30 | 0.38 |

Figure 17: **Comparison of One-D-Piece, ALIT, and KARL with variable token allocation per image satisfying reconstruction $\ell_1 \times 10$ loss $< 0.9$:** At this reconstruction loss threshold, the goal is **not perfect reconstruction**, but rather to evaluate which method achieves acceptable reconstruction quality within the target constraint while using the fewest tokens (i.e., minimizing K̂C). ALIT and KARL use fewer tokens than One-D-Piece for acceptable recons., satisfying $\ell_1 \times 10$ loss $< 0.9$.

| | Input Image | One-D-Piece | ALIT | KARL |

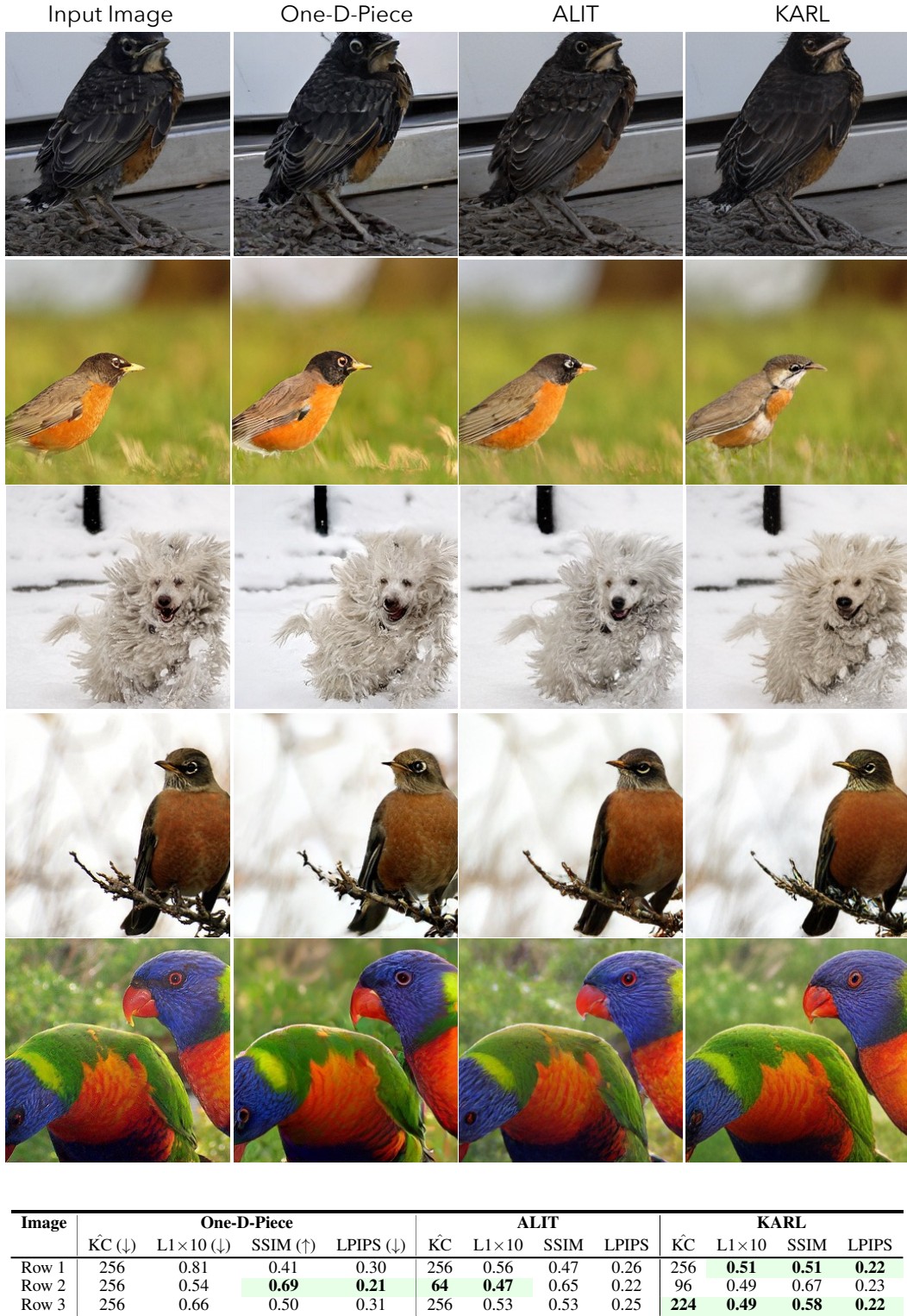

| Image | One-D-Piece | | | | ALIT | | | | KARL | | | |
|-------|-------------|--------------|-----------|-------------|-----|-------|------|-------|-----|-------|------|-------|
| | $\hat{KC}$ (↓) | L1×10 (↓) | SSIM (↑) | LPIPS (↓) | $\hat{KC}$ | L1×10 | SSIM | LPIPS | $\hat{KC}$ | L1×10 | SSIM | LPIPS |
| Row 1 | 256 | 0.81 | 0.41 | 0.30 | 256 | 0.56 | 0.47 | 0.26 | 256 | **0.51** | **0.51** | **0.22** |
| Row 2 | 256 | 0.54 | **0.69** | **0.21** | **64** | **0.47** | 0.65 | 0.22 | 96 | 0.49 | 0.67 | 0.23 |
| Row 3 | 256 | 0.66 | 0.50 | 0.31 | 256 | 0.53 | 0.53 | 0.25 | **224** | **0.49** | **0.58** | **0.22** |
| Row 4 | 256 | 0.64 | 0.64 | 0.26 | **192** | 0.49 | 0.67 | 0.22 | 224 | **0.48** | **0.71** | **0.19** |
| Row 5 | 256 | 0.92 | 0.32 | 0.34 | 256 | 0.69 | 0.35 | 0.30 | 256 | **0.65** | **0.39** | **0.25** |

Figure 18: **Comparison of One-D-Piece, ALIT, and KARL with variable token allocation per image satisfying reconstruction** $\ell_1 \times 10$ **loss** $< 0.5$**:** At this reconstruction loss threshold, the goal becomes much closer to near **perfect reconstruction** while using the fewest tokens ($\hat{KC}$). ALIT and KARL often save a few tokens from the full 256 token budget, satisfying $\ell_1 \times 10$ loss $< 0.5$. Threshold$= 0.05$ can serve as a de-facto threshold for majority of vision tasks.

