# OpenReview forum: "Single-pass Adaptive Image Tokenization for Minimum Program Search"
_NeurIPS.cc/2025/Conference — NeurIPS 2025 poster_

### Official Review · Reviewer_YZ29 · 2025-06-13

**Clarity:** 2
**Significance:** 2
**Originality:** 2
**Rating:** 4
**Confidence:** 3

**Summary:**

The paper proposes a new method to learn a tokenizer that can adjust its output (token) length that depends on the complexity of input data, image specifically. The innovation in this paper is a new training method  that is run after training an initial tokenizer that outputs a probability value for rejecting/keeping a token. The proposed method takes inspiration from algorithmic information theory that states that the optimal program is the shortest possible ("simplest") program for a given input. Empirical results are provided that shows that the proposed method, KARL (acronym), is able to work as well as a recurrent baseline while being a one step method.

**Questions:**

- Please clarify the details of the training algorithm used in the experiments in the main body of the paper

- Please check and ensure that the values in bold are for the best performing method in Table 1. For example, I see on row 2 that KARL is bolded for LPIPS metric but it appears that ALIT's value is lower for 32 tokens.

- In Figure 5, the 2nd plot on the left shows a lower FID compared to the plot on the right(for variable number of tokens). Why are the variable token length methods not leading to similar FID values? I am unsure if the conclusions that small encoder + large decoder is supported in general. Please clarify

**Ethical Concerns:**

["NO or VERY MINOR ethics concerns only"]

**Final Justification:**

I read through other reviewers' and responses carefully and will provide a final rating after discussions with reviewers and AC.

**Limitations:**

Limitations in appendix.

**Quality:**

2

**Strengths And Weaknesses:**

# Strengths

  - Adaptive tokenizers that can adjust its output length based on input complexity is definitely an interesting area of research. See for example reference [19] that proposes a method called ALIT quoted heavily in the paper that was presented recently.

- The method identifies a drawback with recurrent approaches like ALIT in that they are serial in nature. Instead the paper proposes one approach to shift some of this computation to a second training stage and performs one-shot inference

# Weaknesses

- Algorithmic information theory (AIT) is used to inspire the proposed method. However, the connection does not appear to be very concrete. Kolmogorov Complexity (KC) is approximated via token length. The use of token length by itself is sensible while the connecting to AIT/theoretical KC is vague and somewhat confusing to this reader

- In a similar vein to above, the measure of interestingness via \kappa is defined using SSIM which is from image processing literature. The use of AIT/KC here again is vague and may not even be necessary to define this measure.

- The proposed method needs T, T{\prime} and \epsilon but the details of these values used in experiments are missing in the main paper. I skimmed the appendix but was not able to find the information that i needed for T\prime. Please clarity this information in the main body of the paper

---

> ### Author Rebuttal · Authors · 2025-07-31
>
> We appreciate the reviewer’s appreciation on the motivation and novelty of a single-pass adaptive tokenizer. The concerns raised relate to (1) lack of theoretical grounding in Algorithmic Information Theory (AIT), (2) writing clarity. We address both below.
>
> ---
>
> **We revised Section 4** to better articulate the connection between adaptive tokenizers and AIT. We believe earlier writing may have caused confusion, which we have now clarified. Specifically, we include analogies to KC and Solomonoff Induction, particularly the idea of searching for the shortest program.
>
> One key reference, lossy (complexity-constrained) compression has been discussed in seminal work of Kolmogorov Structure Function [1], supporting our theoretical framing.
>
> ### **Section 4: Adaptive Image Tokenization meets AIT**
>
> **Correlation with KC:** KC formalizes the length of the **shortest program** that **reproduces a data** instance. In our setting, we approximate this by treating the token count as a proxy for program length—each token encodes structured information. Minimizing tokens while preserving fidelity aligns with minimizing description length, and thus, KC. Let’s understand this analogy in more detail
>
> Formally, the **“program”** includes both token sequence and decoder. Since the decoder is fixed and shared across samples, its cost can be treated as constant, leaving minimal tokens as shortest program.
>
> Secondly, because exact reproduction is infeasible in deep learning, we define **“reproduction”** as achieving reconstruction within an error threshold (e.g., ℓ₁ or perceptual loss). This yields a tractable notion of approximate KC, the foundation of our work.
>
> This view parallels the seminal Kolmogorov Structure Function (KSF) [1] (see Example II.4), where KSF(x, α) captures the size of smallest set S containing x such that KC(S) ≤ α. Here, α is a complexity budget, and S "explains" x—analogous to finding the shortest token sequence that reconstructs x within a target loss.
>
> **Searching the shortest program**—
> Algorithmic Probability frameworks like Solomonoff Induction (SI) aim to find the shortest program that explains data, assigning higher probability to simpler programs -- by exhaustively enumerating over programs that reproduce the data when decoded by UTM, in increasing order of KC.
>
> Aligning w/ SI, adaptive tokenizers (e.g., ALIT) approximate this search by varying token budgets at test time, incrementally increasing tokens until the reconstruction meets a fidelity threshold. KARL approximates this shortest-program search in a single pass: it predicts the minimal subset of tokens needed to meet a reconstruction constraint avoiding iterative decoding.
>
> ----
> **Next, we revised the Methods section for clarity. We apologize for earlier confusion.**
>
> The overall procedure is inspired by Kolmogorov Complexity principles (like invariance w/ additional resources/tokens, token minimization) and the training prodecure aligns with the seminal Upside-Down RL Framework.
>
> ### **Section 3: Kolmogorov-Approximating Representation Learning**
>
>
> Our goal is to learn representations that are both compressed and predictive, thus: (a) minimize reconstruction loss, (b) minimize representation length.
>
> While existing adaptive tokenizers rely on iterative passes at test time to find the minimal tokens for a desired quality, we ask: Can we predict minimality in a single pass?
>
> We introduce KARL, a loss-conditioned adaptive tokenizer that approximates the image’s minimum description length—the fewest tokens needed for a target reconstruction threshold. At test time, KARL takes an image, token budget, and loss threshold, and outputs only the necessary tokens, masking out the rest.
>
> **Loss-Conditioned Training of KARL:**
> We define the approximate Kolmogorov Complexity (\\( \\hat{\\text{KC}} \\)) of an image \\( x \\) under a reconstruction threshold \\( \\epsilon \\) and a maximum given token budget, \\( T \\) as:
> \\[
> \\hat{\\text{KC}}\_\\epsilon(x, T) = \\min \\left\\{ t \\leq T \\;\\middle|\\; \\mathcal{L}\_{\\text{rec}}(x, \\hat{x}\_t) \\leq \\epsilon \\right\\}
> \\]
> Here, \\( \\hat{x}\_t \\) is the reconstruction from the first \\( t \\) tokens (out of \\( T \\)), and \\( \\mathrm{L}\_{\\text{rec}} \\) is the reconstruction error. Intuitively, \\( \\hat{\\text{KC}}_\\epsilon(x, T) \\) represents the smallest token count sufficient to reconstruct \\( x \\) within error \\( \\epsilon \\), constrained by the budget \\( T \\). This defines a practical, task-specific approximation of Kolmogorov Complexity.
>
>
> To encourage learning such minimal programs, we train KARL under the assumption of **KC invariance**:
> \\[
> \\hat{\\text{KC}}_\\epsilon(x, T) = \\hat{\\text{KC}}_\\epsilon(x, T + \\Delta T)
> \\]
> That is, **increasing the token budget shouldn't change the minimal token count needed for a given \( \\epsilon \)**. Once the model learns the sufficient subset, adding tokens should not alter the representation. This principle guides KARL to mask out unnecessary tokens during training.
>
>
> **Training Procedure:**
> Each training iteration runs the encoder-decoder pipeline twice, reflecting the principle that KC remains invariant once the sufficient token count is found for reconstruction. We now describe the two steps performed in each training iteration:
>
> **1. Estimating Image Complexity:**
> Given an input image \\( \\mathbf{x} \\), the model first attempts near-lossless compression using a token budget \\( T \\), optimizing:
>
> \\[
> \\mathcal{L}\_{\\text{EIC}} = \\mathcal{L}_{\\text{recon}}(\\hat{\\mathbf{x}}\_T, \\mathbf{x}) + \\beta \\, \\mathcal{L}\_{\\text{quant}}(\\mathbf{z}\_T)
> \\]
>
> where \\( \\hat{\\mathbf{x}}_T = \\texttt{Decoder}(\\texttt{Encoder}(\\mathbf{x}, T, \\epsilon = 0)) \\), and \\( \\beta \\) balances quantization. Here, \\( \\epsilon = 0 \\) enforces near-perfect reconstruction. The resulting reconstruction error \\( \\epsilon_0 = \\left| \\hat{\\mathbf{x}}_T - \\mathbf{x} \\right| \\) defines the target quality for the second run. Sampling \\( T \\) randomly each iteration yields a batch of triplets \{image, token budget \\( T \\), target error \\( \\epsilon_0 \\)\} that guide complexity-aware tokenization.
>
>
>
> **2. Learning to Tokenize at Estimated Complexity:**
> In the second run, given the simulated triplet \{\\mathbf{x}, T, \\epsilon\_0 \}, the encoder receives an increased token budget \\( T + \\Delta T \\). Conditioned on \\( \\epsilon_0 \\), it outputs both token embeddings and halting probabilities \\( \\boldsymbol{\\omega} \\in [0,1]^{T + \\Delta T} \\), indicating which tokens to retain.
>
> The model is optimized using:
>
> \\[
> \\mathcal{L}\_{\\text{LTC}} =
> \\mathcal{L}\_{\\text{recon}}(\\hat{\\mathbf{x}}\_{T+\\Delta T}, \\mathbf{x}) + \\beta \\, \\mathcal{L}\_{\\text{quant}}(\\mathbf{z}\_{T+\\Delta T}) + \\lambda \\, \\mathcal{L}\_{\\text{halt}}(\\boldsymbol{\\omega})
> \\]
>
> where \\( \\hat{\\mathbf{x}}\_{T+\\Delta T} = \\texttt{Decoder}(\\texttt{Encoder}(\\mathbf{x}, T+\\Delta T, \\epsilon = \\epsilon\_0)) \\).
> The encoder is trained to achieve the same recon. quality \\( \\epsilon\_0 \\) while identifying which tokens can be masked out. Halted tokens are excluded from self-attention, thus do not contribute to reconstruction.
>
> The halting loss is:
>
> \\[
> \\mathcal{L}\_{\\text{halt}}(\\boldsymbol{\\omega}) =
> \\text{BCE}(\\boldsymbol{\\omega}\_{0:T}, \\mathbf{0}) +
> \\text{BCE}(\\boldsymbol{\\omega}\_{T:T+\\Delta T}, \\mathbf{1})
> \\]
>
> encouraging \\( \\omega_i \\approx 0 \\) for essential tokens \\( i < T \\) and \\( \\omega\_i \\approx 1 \\) for surplus tokens \\( i \\geq T \\).
>
> The full training objective is \\( \\mathcal{L}\_{\\text{EIC}} + \\mathcal{L}\_{\\text{LTC}} \\). This parallels Upside-Down RL, where the first run sets a condition (\\( \\epsilon\_0 \\)) and the second learns to act under it—akin to reframing RL as supervised learning with reward conditioning.
>
> At inference, only second step is used: given a max token budget & a target error \\( \\epsilon \\), KARL outputs both token embeddings & halting probabilities in a single pass.
>
>
> ---
> **Clarification regarding T':** We believe T' was probably not the best choice of symbol; it's replaced with \\( \\Delta T \\) in the text above. As mentioned earlier, in the **Learning to Estimate Complexity** step, for each image in the batch, we *randomly* sample an input token budget \\( T \\), which results in a \\( \\epsilon \\) error due to lossy compression. Then, for each image, we randomly sample \\( \\Delta T \\) (which is the same as T' in the earlier draft). In the **Tokenize at Complexity** step, the encoder—conditioned on the \\( \\epsilon \\) obtained in the previous step—is tasked to use only the minimal number of tokens from the new input budget of \\( T + \\Delta \\). We then train the network to mask out the extra \\( \\Delta T \\) tokens, since \\( T \\) was already sufficient for \\( \\epsilon \\)-bounded reconstruction.
>
> ---
> **Regarding Interestingness and Connections to AIT** -- Interestingness is deeply rooted in AIT. For example, see Scott Aaronson's blog on "The First Law of Complexodynamics" or Peter Bloem’s slides on Sophistication. Measures like sophistication, logical depth, and KSF [1] are all derivative notions based on Kolmogorov Complexity (KC). While we observed early signs that adaptive tokenizers—especially KARL—could estimate sophistication (via the rate of change in reconstruction quality with respect to token count), the main contribution we emphasize is the novel **single-pass adaptive tokenization** and its core analogy with KC.
>
> ---
> **Figure 5 query** — The left plot uses \\( \epsilon = 0 \\), targeting perfect reconstruction, so FID at 256 tokens is lower. In contrast, the right plot shows KARL stopping early with \\( \epsilon > 0 \\), leading to higher FID. We hope this clarifies the difference.
>
> ---
> **Table formatting** — Thank you for pointing this out; we will correct it.
>
> We hope the two revised sections address the reviewer’s concerns.

---

> > ### Comment · Reviewer_YZ29 · 2025-08-04
> >
> > I thank the authors for walking me through an updated explanation of the proposed method, KARL, as well as using algorithmic info. theory to inspire designing an adaptive image tokenization method.
> >
> > > Formally, the “program” includes both token sequence and decoder. Since the decoder is fixed and shared across samples, its cost can be treated as constant, leaving minimal tokens as shortest program.
> >
> > The point above made in the rebuttal suggests that the decoder is fixed (which is fine). But naively I would wonder whether a different decoder might also lead to a better program. In any case, the authors have done a nice job with their method.
> >
> > I am happy to raise the rating of this paper to a 3. I read through other reviewers' and responses carefully and will provide a final rating after discussions with reviewers and AC.

---

> ### Author Response · Authors · 2025-08-04
> **Response to Reviewer YZ29**
>
> We sincerely thank the reviewer for their response and for confirming that their concerns have been addressed.
>
> > The point above made in the rebuttal suggests that the decoder is fixed (which is fine). But naively I would wonder whether a different decoder might also lead to a better program. In any case, the authors have done a nice job with their method.
>
> This is a great insightful question. As noted in the main paper, the decoder is ideally considered part of the program (though amortized across all images). Accordingly, modifying the decoder architecture can overall affect the description length, and thus the approximate Kolmogorov Complexity (KC) of an image under our model. We explore this effect in detail through controlled ablations in Appendix Figures 13 and 14.
>
> That said, consistent with Algorithmic Information Theory—and as discussed around line 315 of the main paper—the invariance theorem suggests that such the decoder change should impact KC estimates by at most an additive constant. This implies that while the absolute KC values may shift, the relative complexity ordering across inputs should remain stable.
>
> Empirically, this is precisely what we observe. As shown in the rightmost plots of Figures 9 and 10, changing the decoder width or depth has minimal effect on the active token count, suggesting that our KC approximation is effectively invariant under decoder variation. Rather the decoder size influenced the reconstruction quality.
>
> We acknowledge that using a much larger decoder (beyond what is feasible in our academic setup) could either potentially reduce the minimal required token count further (as observed in FlexTok paper) or increase the reconstruction quality, but the relative ordering of image complexities should still hold. We hope this clarifies the query.
>
> ---
>
> More broadly, we respectfully note that we have thoroughly addressed all reviewer questions, and we appreciate all the reviewers’ recognition of the paper’s contributions—namely:
>
> - The novel problem formulation: introducing the first single-pass adaptive tokenizer,
> - Innovative Approach: A loss-conditioned training approach inspired by Kolmogorov Complexity principles and drawing parallels with the Upside-Down RL framework,
> - And the strong empirical performance: across both reconstruction and downstream tasks.
>
> Given this, we remain unsure about the rationale behind the current borderline rejection. We would sincerely appreciate any further clarification or suggestions that could help us strengthen the work. We believe the paper offers a meaningful contribution to progressing the scientific study of visual tokenizers, and we are committed to improving it further based on the community’s feedback.
>
> Thanks again,
> Best Regards!

---

> > ### Comment · Reviewer_YZ29 · 2025-08-07
> >
> > > Given this, we remain unsure about the rationale behind the current borderline rejection. We would sincerely appreciate any further clarification or suggestions that could help us strengthen the work. We believe the paper offers a meaningful contribution to progressing the scientific study of visual tokenizers, and we are committed to improving it further based on the community’s feedback.
> >
> > My impression is that the points raised during the rebuttal requires a significant rewrite of the paper. I will need time to carefully read through all of  responses made above. I sincerely thank the authors for their hard work. I will work with the other reviewers and the AC (as needed) to finalize my recommendation.

---

> ### Author Response · Authors · 2025-08-07
> **Response to Reviewer YZ29**
>
> Thank you again for your response!
>
> Just to highlight, we mainly revised Sections 3 and 4 to make them easier to read and to better emphasize the key components (as thoughtfully suggested by the reviewer). The overall content remains the same or very similar to the original main paper. Our latest version, based on the reviewer's feedback, is already ready and encapsulates all the contents of the rebuttal.
>
> ---
>
> We truly appreciate the reviewer's effort in engaging with other reviewers and the Area Chair to arrive at the final rating. As mentioned before, we believe the paper makes significant contributions to the scientific (and still relatively fresh) study of visual tokenizers. We hope these contributions are recognized and not overshadowed or scooped.
>
> We trust that the main paper, along with the rebuttal—which addresses all raised concerns—will be given a fair and thorough evaluation.
>
> Best Regards!

---

### Official Review · Reviewer_HAzc · 2025-06-29

**Clarity:** 1
**Significance:** 3
**Originality:** 3
**Rating:** 4
**Confidence:** 1

**Summary:**

This paper introduces KARL, a single-pass adaptive image tokenizer that aims to predict the optimal number of tokens for image representation without requiring iterative search at inference time. The authors draw connections between adaptive tokenization and Algorithmic Information Theory (AIT), proposing a loss-conditioned training strategy that learns to halt token allocation once sufficient representational capacity is reached. The method is evaluated and compared against existing approaches like ALIT and One-D-Piece.

**Questions:**

- How sensitive is the method to the choice of reconstruction loss thresholds during training?

**Ethical Concerns:**

["NO or VERY MINOR ethics concerns only"]

**Final Justification:**

Thanks for the reply from the authors. I would like to raise the rating to 4.

**Limitations:**

- No analysis of failure cases or method limitations.
- Lack of theoretical guarantees about optimality of learned token counts.

**Quality:**

2

**Strengths And Weaknesses:**

Strengths:
- **Practical Contribution**: The single-pass approach addresses a real computational bottleneck in existing adaptive tokenization methods that require iterative search at inference time.
- **Scaling Law Analysis**: The systematic study of various architectural choices (encoder/decoder size, continuous vs. discrete tokenization) provides useful insights.

Weaknesses:
- **Weak Theoretical Foundation**: The connection to Algorithmic Information Theory is superficial and lacks a rigorous mathematical formulation. Section 4 reads more like a philosophical discussion than a technical contribution.  This significantly undermines the claimed theoretical contribution. Additionally, the definition of "interestingness" and its claimed connection to sophistication and logical depth appear ad-hoc and lack theoretical justification.
- **Misplaced Content**: A substantial portion of Section 4 discussing AIT concepts, sophistication, and logical depth should be moved to related work. The current structure makes the paper feel disorganized.
- **Missing Complexity Analysis**: Despite claiming efficiency advantages over iterative methods, the paper provides no computational complexity analysis comparing training time, memory requirements, or actual inference speedup against baseline methods like ALIT and One-D-Piece.
- **Limited Theoretical Novelty**: The core technical contribution is essentially a masking strategy during training. While practical, this represents an incremental engineering improvement rather than a fundamental advance in adaptive tokenization.
- **Evaluation Concerns**: (1) The variable token allocation comparison may not be entirely fair since it's the primary design goal of KARL. (2) No statistical significance testing due to computational constraints.

---

> ### Author Rebuttal · Authors · 2025-07-31
>
> We appreciate the reviewer’s appreciation on the practicality of single-pass adaptive tokenization and our system study of scaling laws.
>
> ---
>
> We revised Section 4 to better articulate the connection between adaptive tokenizers and AIT. We believe earlier writing may have caused confusion, which we have now clarified. Specifically, we include analogies to KC and Solomonoff Induction, particularly the idea of searching for the shortest program.
>
> One key reference, lossy (complexity-constrained) compression has been discussed in seminal work of Kolmogorov Structure Function [1], supporting our theoretical framing.
>
> ### **Section 4: Adaptive Image Tokenization meets AIT**
>
> **Correlation with KC:** KC formalizes the length of the **shortest program** that **reproduces a data** instance. In our setting, we approximate this by treating the token count as a proxy for program length—each token encodes structured information. Minimizing tokens while preserving fidelity aligns with minimizing description length, and thus, KC. Let’s understand this analogy in more detail
>
> Formally, the **“program”** includes both token sequence and decoder. Since the decoder is fixed and shared across samples, its cost can be treated as constant, leaving minimal tokens as shortest program.
>
> Secondly, because exact reproduction is infeasible in deep learning, we define **“reproduction”** as achieving reconstruction within an error threshold (e.g., ℓ₁ or perceptual loss). This yields a tractable notion of approximate KC, the foundation of our work.
>
> This view parallels the **seminal Kolmogorov Structure Function (KSF) [1] (see Example II.4)**, where KSF(x, α) captures the size of smallest set S containing x such that KC(S) ≤ α. Here, α is a complexity budget, and S "explains" x—analogous to finding the shortest token sequence that reconstructs x within a target loss.
>
> **Searching the shortest program**—
> Algorithmic Probability frameworks like Solomonoff Induction (SI) aim to find the shortest program that explains data, assigning higher probability to simpler programs -- by exhaustively enumerating over programs that reproduce the data when decoded by UTM, in increasing order of KC.
>
> Aligning w/ SI, adaptive tokenizers (e.g., ALIT) approximate this search by varying token budgets at test time, incrementally increasing tokens until the reconstruction meets a fidelity threshold. KARL approximates this shortest-program search in a single pass: it predicts the minimal subset of tokens needed to meet a reconstruction constraint avoiding iterative decoding.
>
> We have exciting general response regarding the alignment of adaptive visual tokenization and algorithmic information theory, and hope the writing would clarify reviewer's concern regarding the same.
>
> ---
>
> **Lack of theoretical guarantees about optimality of learned token counts**
>
> We respectfully clarify that computing the **theoretically optimal number of tokens** per data point is inherently intractable—this is a known limitation of Kolmogorov Complexity (KC) and Solomonoff Induction (SI), not specific to our method. Our work aims to **approximate** KC by treating the minimal number of tokens (under a learned decoder) as a proxy for the shortest program that reproduces an image—an approach, to our knowledge, not previously explored for visual tokenization.
>
> A useful theoretical framing of KARL (like all auto-encoders) is via classical rate-distortion analysis:
>
> \\[
> Z_{\\text{optimal}} = \\min_Z \\left[ -H(X \\mid Z) + \\lambda D(X, \\hat{X}(Z)) \\right]
> \\]
>
> - \\( H(X \\mid Z) \\): conditional entropy—uncertainty in \\( X \\) given \\( Z \\).
> - \\( D(X, \\hat{X}(Z)) \\): reconstruction loss; \\( \\lambda \\) balances compression vs. fidelity.
> ---
>
> **Limited Theoretical Novelty**
>
> While the high-level idea—masking out excess tokens given a token budget—may seem straightforward, the proposed method introduces several non-trivial innovations that, in our view, contribute meaningfully to the field.
>
> **Theoretical grounding in Kolmogorov Complexity (KC):**
> Our approach is not merely an engineering trick but is guided by foundational principles from Algorithmic Information Theory. In particular, we leverage the invariance property of KC—that KC is an intrinsic property of data and should remain unchanged even when more computational resources (i.e., tokens) are available. This motivates our learning objective: once a minimal token subset sufficient to reconstruct an image is identified, additional tokens should be deemed unnecessary.
>
> **Training innovation via loss-conditioned supervision:**
> Naively applying masking during training is non-differentiable and, in our experiments, fails to converge. Common strategies like straight-through estimation or reinforcement learning introduce complexity and instability. In contrast, our method draws inspiration from the Upside-Down Reinforcement Learning framework, enabling a stable two-step procedure within each training iteration:
> (a) Estimate Complexity — mapping the image to a randomly selected token budget and observing the resulting reconstruction error \\( \\epsilon \\);
> (b) Tokenize at Learned Complexity — using \\( \\epsilon \\) as a conditioning signal to guide the encoder to mask out redundant tokens from an expanded budget. This enables effective one-shot prediction of a minimal token set for a given reconstruction constraint.
>
> Overall, we believe that combining this theoretical motivation with a stable, practical training strategy for single-pass adaptive tokenization represents a meaningful technical contribution. We are encouraged by the broader response, which found the method novel and interesting, especially in the context of visual tokenization.
>
> We hope this response answers the concern of technical novelty. We apologize it this wasn't clear upfront and we will improve the writing.
>
> ---
> **Evaluation Concerns**
> We want to clarify that variable token allocation is a central goal of all adaptive tokenizers—not just KARL. For example, ALIT explicitly emphasizes this (Figure 6 of their paper). The key distinction lies in **how** adaptation is achieved.
>
> KARL predicts when to halt tokenization in a single forward pass, conditioned on a reconstruction threshold. In contrast, methods like ALIT rely on test-time search with multiple encoder-decoder passes, using decoder feedback to iteratively minimize token count. While this allows slightly tighter token estimates, it comes with higher FLOPs and latency.
>
> As a result, ALIT achieves a marginally lower estimated \\( \\hat{KC} \\) than KARL (see Table 2), but at the cost of increased compute. These approaches are complementary and could be combined—for instance, using KARL’s prediction as an efficient initialization for further refinement.
>
> ---
> **Limitations**
>
> - FID gap despite strong per-image metrics: KARL performs competitively or better than existing tokenizers on all metrics except the distribution-level metric FID. This may be due to KC-guided training favoring the search for the shortest program per image—optimizing for single-image metrics (LPIPS, SSIM, PSNR, DreamSim)—rather than global distribution alignment. A deeper analysis of this trade-off would be valuable.
>
> - Inability of KC to distinguish noise from structure: As noted in the paper, Kolmogorov Complexity assigns high complexity to both noisy and highly structured images. Higher-order measures such as Sophistication or Logical Depth can make this distinction and form the basis of our proposal for interestingness. However, our current interestingness formulation is a simple delta on KC, not a learned signal.
>
> - General limitation of all adaptive tokenizers: Current methods, including KARL, are constrained to a maximum token budget per image (e.g., 256 tokens for 256×256 images), limiting granularity.
>
> **Future Work**
> Exploring the use of learned compressed tokens to train downstream models—such as vision-language models or video models—is a promising and timely direction for further research.
>
> ---
> **Misplaced Content:** Thank you for the suggestion. In the latest revision, Section 4 focuses only on the connections to Kolmogorov Complexity and Solomonoff Induction. We will move broader background and references to AIT concepts to the Related Work section.
>
> ---
> **Complexity Analysis**
>
> We thank the reviwer for bringing this up. We compared computational complexity in terms of the number of encoder/decoder calls in Table 2, which is architecture-agnostic and avoids confounds like GPU type or precision. For fair comparison, all baselines use the same encoder/decoder architecture.
>
> Here are the requested numbers in terms of Glops and executing time on a single H100 GPU with FP32 precision:
>
> - **Encoder GFlops:** KARL ≈ One-D-Piece ≈ First iteration of ALIT ≈ 80 GFlops
> - **ALIT:** Requires 8 iterations during training, and ~4 iterations at test time for \\( \epsilon = 0.9 \\)-bounded reconstruction. Each additional ALIT iteration adds ~30 GFlops.
> - **Decoder GFlops:** ALIT and One-D-Piece require multiple decoding passes, each ≈ 80 GFlops.
>
> Wall-clock Time (ms):
> - First encoder or decoder pass: ~7 ms
> - Each additional ALIT encoder iteration: ~4 ms
>
> ---
> We hope we have addressed all concerns—through improved writing, citing the strong reference on Kolmogorov Structure Function (KSF), and highlighting the strong analogy between AIT and adaptive tokenization. Additionally, we clarified how our training strategy emerges naturally from KC principles and aligns with the seminal Upside-Down RL framework, which we believe helps address any concerns regarding technical novelty.
>
> Interestingly, we’ve also heard others point out conceptual parallels between our work and the recent interesting H-Net paper on dynamic tokenization for LLMs—both approaches learn *when to stop* tokenization.
> We’re grateful for your review and are happy to answer any further questions.
>
> [1] Kolmogorov’s Structure Functions and Model Selection

---

> > ### Comment · Reviewer_HAzc · 2025-08-06
> >
> > The author's reply solved most of my doubts. However, I think the author should conduct more extensive research on the baseline methods that their method needs to be compared with, and explain the differences and advantages between them.

---

> ### Author Response · Authors · 2025-08-06
> **Response to Reviewer HAzc**
>
> We sincerely thank the reviewer for their response and for confirming that their concerns have been addressed.
>
> ---
>
> We made a **figure highlighting the taxonomy of recent adaptive tokenizers in our latest version**, however due to neurips restrictions on rebuttal document, we couldn't include it in the rebuttal. Here is a textual description of that figure --
>
> - Adaptive tokenizers can be bucketed into two broad categories (as mentioned in the paper), namely **Matryoshka-style (ElasticTok, FlexTok, One-D-Piece) and Recurrent Tokenizers (ALIT)**.
> - **Matryoshka-based Tokenizers**
>     - ❌ Smaller embeddings are constrained as subsets of the larger embedding & **hence adding more tokens only refines the initial ones, rather than discovery of new concepts as in ALIT and KARL (for example compare One-D-Piece Figure 1 with ALIT Figure 16). KARL showcases similar properties as ALIT.**
>     - ❌ Not aligned with the concept of Prefix Kolmogorov Complexity (a key concept in AIT, Universal Turing Machine -- which states a shortest program need not be subset of other programs).
>     - ❌ Requires multiple Decoder runs to identify smallest #tokens.
> - **Recurrent Tokenizers**
>     - ✅ Kolmogorov Complexity aligned, shortest program or minimal tokens can be found by searching over multiple recurrent iterations.
>     - ❌ Requires multiple Encoder & Decoder runs to identify smallest #tokens
> - **KARL:**
>     - ✅ Kolmogorov Complexity aligned & inspired.
>     - ✅ Single-pass.
>
> We hope this textual description solves your query of **explain the differences and advantages between them**. We will include the visual version of this taxonomy in the final camera ready version.
>
> ---
>
> **More information regarding choice of these baselines**: We compared against two recent strong state-of-the-art baselines—ALIT (ICLR 2025) and One-D-Piece (2025). Other notable works include Flextok, ElasticTok, (non-adaptive) FlowMo, Titok, and older methods like VAE and VQGAN.
>
> - FlexTok, One-D-Piece and ElasticTok fall under the same Matryoshka-style tokenizers, and since ALIT, One-D-Piece, and Titok all adopt a similar 2-stage GAN-based training pipeline with 2D VQGAN/VAE tokens, we chose One-D-Piece for a fair, apples-to-apples comparison with KARL focusing on our core contributions.
>
> - Furthermore, Flextok models are much large-scale (e.g., trained with REPA-style losses using DINO, which is trained on much broader datasets), making direct comparisons with KARL, ALIT, and One-D-Piece less fair in scope. That said, we can include Flextok results in the final version.
>
> - Numbers for VAE, VQGAN, Titok are same as mentioned in the ALIT paper, and can be simply copied from the ALIT paper.
>
> **To the best of our knowledge, these represent the latest adaptive tokenization baselines. If we’ve missed the specific one you're referring to, we’d be very grateful if you could point us to it directly—we’d be happy to evaluate it, either within the remaining time or for the final version.**
>
> We hope this helps clarify things, and we’d really appreciate a quick response—especially with the discussion period nearing its end.

---

> > ### Author Response · Authors · 2025-08-06
> > **Comparisons with new baseline – Flextok**
> >
> > Based on the reviewer’s request, we have now included an evaluation of the FlexTok model. This comparison was conducted shortly after receiving the request and builds upon our already comprehensive baseline coverage and fair evaluation methodology.
> >
> > To ensure a meaningful comparison, we cloned the official FlexTok repository and evaluated their smallest available model, which includes 12 encoder and 12 decoder layers—significantly larger than our architecture, which uses only 8 encoder and 8 decoder layers. All experiments were conducted on the ImageNet100 dataset for consistency.
> >
> > \\[
> > \\begin{array}{|c|cc|cc|cc|cc|}
> > \\hline
> > \\textbf{Method}
> > & \\text{L1 (32)} & \\text{FID (32)}
> > & \\text{L1 (64)} & \\text{FID (64)}
> > & \\text{L1 (128)} & \\text{FID (128)}
> > & \\text{L1 (256)} & \\text{FID (256)} \\\\
> > \\hline
> > \\text{KARL-vae-quant-1D}
> > & \\textbf{1.141} & 33.70
> > & \\textbf{0.976} & 22.36
> > & \\textbf{0.839} & \\textbf{14.95}
> > & \\textbf{0.701} & \\textbf{9.74} \\\\
> > \\text{Flextok-vae-quant-1D}
> > & 1.749 & \\textbf{25.28}
> > & 1.505 & \\textbf{21.95}
> > & 1.198 & 19.58
> > & 0.982 & 17.26 \\\\
> > \\hline
> > \\end{array}
> > \\]
> >
> > \\[
> > \\begin{array}{|c|cc|cc|cc|cc|}
> > \\hline
> > \\textbf{Method}
> > & \\text{SSIM (32)} & \\text{LPIPS (32)}
> > & \\text{SSIM (64)} & \\text{LPIPS (64)}
> > & \\text{SSIM (128)} & \\text{LPIPS (128)}
> > & \\text{SSIM (256)} & \\text{LPIPS (256)} \\\\
> > \\hline
> > \\text{KARL-vae-quant-1D}
> > & \\textbf{0.3334} & \\textbf{0.4014}
> > & \\textbf{0.3758} & \\textbf{0.3487}
> > & \\textbf{0.4328} & \\textbf{0.2905}
> > & \\textbf{0.5023} & \\textbf{0.2338} \\\\
> > \\text{Flextok-vae-quant-1D}
> > & 0.2334 & 0.5259
> > & 0.2659 & 0.4768
> > & 0.3333 & 0.4165
> > & 0.4024 & 0.3607 \\\\
> > \\hline
> > \\end{array}
> > \\]
> >
> > As shown in the new results table above, **KARL consistently outperforms FlexTok in all single-image reconstruction quality and performs comparably on distribution-wide FID metric. In addition to improved performance, KARL also offers substantial efficiency advantages: while FlexTok relies on iterative latent sampling, KARL operates in a single forward pass, making it significantly faster.** Note: we have also thoroughly ablated KARL with both vae, vqgam, continuous ve discrete tokenizers in appendix scaling law plot.
> >
> > We note that FlexTok represents an industry-level, state-of-the-art model trained with DINO supervision. That KARL matches or surpasses its performance—despite not being trained with that supervision and more efficient—underscores the strength of our approach. Combined with favorable comparisons to other baselines such as One-D-Piece, ALiT, VAE, and VQGAN (as reported in the ALiT paper), this new result further supports the significance and competitiveness of KARL.
> >
> > We hope that our responses—both the detailed taxonomy positioning KARL among adaptive tokenizers (in previous response) and the new experimental comparison with FlexTok—satisfactorily address the reviewer’s concerns.
> >
> > **Given that all reviewers have already acknowledged the novelty and significance of our approach, we respectfully hope that the reviewer may consider updating their score to better reflect the contributions and strengths of the paper.**
> >
> > Best Regards!

---

> ### Author Response · Authors · 2025-08-08
> **Seeking Reviewer Clarification**
>
> We thank the reviewer again for their earlier response and for **acknowledging that all previous queries have been resolved (within the NeurIPS rebuttal limit)**
>
> **Following your response on August 6th, we promptly evaluated an additional baseline and introduced a detailed taxonomy of adaptive tokenizers**—examining them from both practical and AIT perspectives—which, to the best of our knowledge, has not been presented before. Drawing on our experience with this topic since its early works, we have also incorporated nearly all state-of-the-art adaptive tokenization baselines, comparing them across a wide range of metrics in a fair and consistent manner.
>
> We would be grateful if the reviewer could clarify the remaining reasons for recommending rejection, as based on the rebuttal response we are not aware of any outstanding concerns. We genuinely believe—also supported by broader feedback—that the paper offers multiple strong contributions and advances the field of visual tokenization in a fresh and meaningful way. We remain committed to improving the paper, and it would be unfortunate if its contributions were overlooked without further clarification.
>
> Best Regards!

---

### Official Review · Reviewer_95ih · 2025-07-03

**Clarity:** 3
**Significance:** 3
**Originality:** 3
**Rating:** 4
**Confidence:** 3

**Summary:**

This paper proposes KARL, a one-pass adaptive tokenizer that predicts the minimal number of image tokens needed for reconstruction, inspired by Kolmogorov Complexity. Unlike prior methods that require iterative search, KARL learns to halt token allocation using reconstruction loss as a signal. It achieves comparable or better performance to existing methods while being more efficient. The authors also connect adaptive tokenization to concepts from Algorithmic Information Theory and introduce an image interestingness score aligned with human judgments.

**Questions:**

How sensitive is KARL’s performance to the choice of reconstruction loss threshold during training and inference? Would different thresholds lead to significantly different token allocations or interestingness scores?

**Ethical Concerns:**

["NO or VERY MINOR ethics concerns only"]

**Final Justification:**

Thanks for the authors' response, and they have addressed most of my concerns - I've raised the score accordingly.

**Limitations:**

Yes

**Quality:**

2

**Strengths And Weaknesses:**

S1: KARL avoids expensive iterative search at test time by learning to allocate tokens in a single forward pass. The number of tokens used per image serves as an intuitive proxy for complexity, aligning with human perception.

S2: The method matches or outperforms baselines on standard image reconstruction metrics. The work draws thoughtful connections between adaptive tokenization and principles from Algorithmic Information Theory and introduced interpretable metrics.

W1: The experiments focus primarily on image reconstruction; it's unclear how well KARL generalizes to other vision tasks.

W2: The paper introduces multiple novelty ideas, but doesn’t ablate their individual contributions or alternatives.

W3: While the paper mentions distribution shifts, there's minimal quantitative analysis of how well the method handles out-of-distribution images.

W4: (minor) The paper lacks error bars which makes it hard to assess the robustness of the performance.

---

> ### Author Rebuttal · Authors · 2025-07-31
>
> ### **Evaluation on Downstream Tasks**
>
> Thank you for the great suggestion. We have now thoroughly evaluated KARL on multiple downstream tasks—CLIP alignment, depth estimation, semantic segmentation, vision-language modeling (VLM), and linear probing for classification. Our findings show that KARL performs comparably to other tokenizers while operating in a single pass. Additional insights and detailed results are provided in our response to Reviewer 2QbU.
>
> ---
>
> ### **Quantitative Analysis on OOD Datasets**
>
> Thank you for the great suggestion. We evaluated the reconstruction performance of KARL, ALIT, and One-D-Piece on three datasets: ImageNet (in-distribution), COCO (moderately OOD), and Wikipedia Images (highly OOD). This revealed an interesting trend:
>
> * KARL's relative performance gap improves as we move further OOD—from ImageNet to COCO to Wikipedia—suggesting that training for minimality may yield more generalizable abstractions, an idea well-grounded in Algorithmic Information Theory (AIT).
>
> **Trained on Imagenet, Evaluated on COCO**
> \\[
> \\begin{array}{|c|ccc|ccc|ccc|}
> \\hline
> \\textbf{Method} & \\text{L1} \\times \\text{10 (32)↓} & \\text{LPIPS (32)↓} & \\text{SSIM (32)↑} & \\text{L1} \\times \\text{10 (64)↓} & \\text{LPIPS (64)↓} & \\text{SSIM (64)↑} & \\text{L1} \\times \\text{10 (256)↓} & \\text{LPIPS (256)↓} & \\text{SSIM (256)↑} \\\\
> \\hline
> \\text{One-D-Piece} & 1.372 & 0.482 & 0.333 & 1.178 & 0.414 & 0.356 & 0.945 & 0.326 & 0.407 \\\\
> \\text{ALIT} & 1.163 & 0.419 & 0.339 & 0.984 & 0.357 & 0.383 & 0.737 & 0.272 & 0.464 \\\\
> \\text{KARL} & \\textbf{1.112} & \\textbf{0.405} & \\textbf{0.358} & \\textbf{0.956} & \\textbf{0.349} & \\textbf{0.399} & \\textbf{0.636} & \\textbf{0.228} & \\textbf{0.527} \\\\
> \\hline
> \\end{array}
> \\]
>
> **Trained on Imagenet, Evaluated on Wikipedia Image Dataset (OOD)**
>
> \\[
> \\begin{array}{|c|ccc|ccc|ccc|}
> \\hline
> \\textbf{Method} & \\text{L1} \times \\text{10 (32)↓} & \\text{LPIPS (32)↓} & \\text{SSIM (32)↑} & \\text{L1} \times \\text{10 (64)↓} & \\text{LPIPS (64)↓} & \\text{SSIM (64)↑} & \\text{L1} \times \\text{10 (256)↓} & \\text{LPIPS (256)↓} & \\text{SSIM (256)↑} \\\\
> \\hline
> \\text{One-D-Piece} & 1.125 & 0.3122 & 0.4450 & 0.956 & 0.2390 & 0.4725 & 0.747 & 0.1617 & 0.5247 \\\\
> \\text{ALIT} & 0.915 & 0.3674 & 0.4638 & 0.767 & 0.3121 & 0.5063 & 0.567 & 0.2394 & 0.5835 \\\\
> \\text{KARL} & \\textbf{0.863} & \\textbf{0.2507} & \\textbf{0.4915} & \\textbf{0.733} & \\textbf{0.1950} & \\textbf{0.5325} & \\textbf{0.485} & \\textbf{0.1206} & \\textbf{0.6465} \\\\
> \\hline
> \\end{array}
> \\]
>
>
>
> ---
>
> ### **More Ablations**
>
> As acknowleged, we conducted extensive ablations on architectural choices and their effect on learned KC and minimal token counts. This includes variations in encoder/decoder depth and width, continuous vs. discrete tokenization, and quantization codebook size. Thanks to your suggestion, we’ve now also included evaluations on downstream tasks and IID vs. OOD settings. We’re happy to explore additional experiments based on further feedback from the reviewer.
>
> ---
>
> ### **Choice of Reconstruction Thresholds During Training and Inference**
>
> A key strength of KARL is its principled treatment of reconstruction thresholds. During training, each image is either trained with \\( \epsilon = 0.0 \\) (lossless) or with a threshold dynamically set by the previous iteration’s reconstruction quality—effectively aligning token allocation with image complexity. This self-regulating mechanism removes the need for manual threshold tuning, a subtle but powerful feature of KARL's training design that we plan to explore further for more tasks, modalities. Basically, each image is trained for its own inherent comoplexity (or value of epsilon).
>
> At inference time, the threshold \\( \epsilon \\) can be chosen based on the needs of downstream tasks. For example, as shown in our downstream evaluations (see response to Reviewer 2QbU), coarse tasks like CLIP alignment tolerate higher \\( \epsilon \\) (e.g., 0.5–0.9) with minimal performance drop, while yielding significant token savings.
>
> ---
>
> ### **Missing Error Bars**
>
> We assume the reviewer refers to error bars across seeds and other sources of randomness, including statistical significance testing. This is a valuable suggestion—we will include such analyses in the final version.
>
> ---
>
> We hope our overall contributions—including the novel problem statement of single-pass adaptive tokenization, the loss-conditioned training strategy (inspired by the seminal Upside-Down Reinforcement Learning framework), theoretical grounding in Kolmogorov Complexity and AIT, extensive scaling law analysis, and now, comprehensive downstream evaluations—help address all of your concerns.

---

> > ### Author Response · Authors · 2025-08-07
> > **Follow up**
> >
> > Thank you again for your review. We believe we’ve addressed all your queries and appreciate your feedback—it has helped us make the paper more complete. As the discussion period is coming to a close, we would be grateful for any follow up comments and rating update. We remain committed to improving the draft based on any additional input.
> >
> > As shown in our rebuttal, KARL performs on par with state-of-the-art models on downstream evaluation tasks. Furthermore, the OOD dataset evaluations reveal an interesting trend: as the data distribution shifts further out-of-distribution, KARL outperforms prior methods—ALiT and One-D-Piece—by a wider margin than on IID datasets. We thank the reviewer for this valuable suggestion.
> >
> > Finally, we believe the paper contributes meaningfully to the scientific study of visual tokenizers, with several novel and important elements. We trust that the main submission, along with the rebuttal—which addresses all raised concerns—will receive a fair and thorough evaluation.

---

### Official Review · Reviewer_2QbU · 2025-07-03

**Clarity:** 3
**Significance:** 1
**Originality:** 3
**Rating:** 4
**Confidence:** 3

**Summary:**

This paper introduces a novel one-shot adaptive tokenizer named KARL, designed for vision representation learning. Inspired by principles from Algorithmic Information Theory (AIT), particularly Kolmogorov Complexity, the method aims to allocate variable-length token representations that reflect the intrinsic complexity of each image. Unlike standard approaches that assign fixed-length representations or adaptive methods that rely on test-time search, KARL predicts the appropriate number of tokens for an image in a single forward pass, stopping when an approximation of its Kolmogorov Complexity is reached. The authors demonstrate that KARL achieves competitive performance with recent adaptive tokenization methods while offering greater efficiency. In addition to empirical evaluations, the paper presents a conceptual study linking adaptive tokenization to AIT concepts such as sophistication and logical depth, and shows that KARL’s predictions correlate well with human judgments of image complexity and interestingness.

**Questions:**

1. **Justify the Use of Image Reconstruction as the Sole Evaluation Task**
Given that the goal is to learn meaningful, adaptive representations, why is image reconstruction chosen as the primary (and only) evaluation task? In recent literature, reconstruction is often seen as a proxy task and not a reliable indicator of representation quality for downstream applications. Please clarify why this choice was made, and to what extent you believe it reflects the utility of the learned representations. Adding even a few representative downstream tasks (e.g., classification, segmentation, or vision-language tasks) would significantly strengthen the empirical validation.

2. **Evaluate Generalization to Downstream Tasks**
To demonstrate the broader utility of KARL, it would be helpful to test the learned representations on common downstream vision or multimodal tasks. For example, how well do KARL’s token representations perform when used as input to image classification, retrieval, or visual question answering models? If time constraints prevent such additions in this version, please provide a detailed plan or preliminary results, and discuss potential challenges in extending KARL to these settings.

3. **Broaden the Comparison with Other Adaptive Methods**
Currently, the comparison with existing adaptive tokenization methods is limited. Could the authors include more recent or diverse baselines, especially those that have been evaluated on vision tasks? A more comprehensive empirical comparison would help position KARL more clearly in the current landscape and clarify its advantages in terms of both performance and efficiency.

**Ethical Concerns:**

["NO or VERY MINOR ethics concerns only"]

**Final Justification:**

After the authors' rebuttal, my most concerns are solved. And I decide to raise my rating to 4.

**Limitations:**

The authors have not adequately addressed the limitations or potential negative societal impacts of their work. While the method is theoretically motivated and technically sound, the paper does not include a discussion of its limitations—such as the fact that it is only validated on image reconstruction tasks, which may not generalize to practical vision or multimodal applications.

**Paper Formatting Concerns:**

No.

**Quality:**

3

**Strengths And Weaknesses:**

## Strengths

* The paper is clearly written and easy to follow, with well-structured explanations and motivation.

* The approach is grounded in a solid theoretical perspective, drawing from Algorithmic Information Theory (AIT) to inform the design of a dynamic tokenization strategy for vision models.

* The idea of linking token count to Kolmogorov Complexity is novel and conceptually insightful.

## Weaknesses

* The method is only evaluated on image reconstruction tasks, which are not widely considered strong indicators of representation quality in current vision or multimodal learning research.

* To better validate the effectiveness of the proposed approach, it would be important to test it on downstream tasks, such as image classification, object detection, or vision-language tasks (e.g., VQA).

* Without such downstream evaluations, the current experiments offer limited evidence for the practical utility or generalization ability of the method.

---

> ### Author Rebuttal · Authors · 2025-07-31
>
> ### **Evaluating Generalization to Downstream Tasks**
>
> We thank the reviewer for the great suggestion regarding downstream evaluation. We have now conducted a comprehensive set of evaluations. Furthermore, we evaluated both fixed token allocation per image and variable token allocation settings. We are excited to share our findings—
>
> * ALIT is the strongest adaptive tokenizer across most metrics, using a recurrent transformer that incrementally adds tokens—each step incurring extra FLOPs. KARL generally consistently ranks second, followed by the matryoshka-style One-D-Piece. **Notably, all three perform comparably, and KARL matches this performance in a single pass**, highlighting its promise as a effecient, scalable alternative. A compelling future direction is to combine KARL with ALIT-style recurrent refinement—refining minimal representation with more iterative loops.
>
> * In the tables below, the 256-token column corresponds to the setting where the input token budget is 256 and \\( \epsilon = 0.0 \\), meaning all tokens are used for lossless reconstruction. As shown across CLIP, Depth, and Semantic Segmentation metrics, the performance drop from \\( \epsilon = 0.0 \\) to \\( \epsilon = 0.03 \\) and \\( \epsilon = 0.05 \\) is minimal—highlighting the benefit of KARL: allowing slight reconstruction loss has little to no impact on downstream performance. As expected, the effect of increasing \\( \epsilon \\) is especially marginal for coarse tasks like CLIP alignment.
>
> * Note: Compared to KARL and ALIT, One-D-Piece requires 30+ more tokens to achieve \\( \epsilon = 0.09 \\)-bounded reconstructions with similar downstream performance (see Table 2 in the main paper).
>
> Overall, the metrics consistently highlight the significance of KARL—achieving minimal or no drop in performance while operating in a single pass. We hope this addresses the reviewer’s concerns regarding downstream task evaluations, and we would be happy to explore additional tasks upon suggestion.
>
>
> **Evaluating on CLIP Text-Image Downstream Task**
>
> | Approach      | 32 tokens | 64 tokens | 256 tokens | ε = 0.03 | ε = 0.05 | ε = 0.09 |
> |---------------|-------------------------------|-------------------------------|---------------------------------|-----------------------|-----------------------|-----------------------|
> | One-D-Piece   | 82.1                          | 86.7                          | 91.5                            | 91.5                | 89.8                | 89.6                |
> | ALIT          | **84.6**                      | **88.5**                      | **92.8**                        | **92.7**              | **92.2**              | **89.6**              |
> | KARL          | 82.3                          | 86.7                          | 91.2                            | 91.7                 | 90.7                 | 87.1                 |
>
>
> **Evaluating on Depth Estimation Downstream Task**
>
> | Approach      | 32 tokens | 64 tokens | 256 tokens, ε = 0.0 | 256 tokens, ε = 0.03 | 256 tokens, ε = 0.05 | 256 tokens, ε = 0.09 |
> |---------------|---------------------------|----------------------------|-----------------------------|---------------------------|---------------------------|---------------------------|
> | One-D-Piece   | 2.73                      | 2.11                       | 1.53                        | 1.53                      | 1.55                      | 1.77                      |
> | ALIT          | **2.04**                      | **1.62**                   | 1.17                        | 1.18                      | **1.24**                    | **1.51**                  |
> | KARL          | 2.19                  | 1.74                       | **1.16**                    | **1.18**                  | 1.32                  | 1.76                      |
>
>
> **Evaluating on Semantic Segmenatation**
>
>
> | Approach      | F-measure @ 32 | F-measure @ 64 | F-measure @ 256 | F-measure (ε = 0.03) | F-measure (ε = 0.05) | F-measure (ε = 0.09) |
> |---------------|----------------|----------------|------------------|------------------------|------------------------|------------------------|
> | One-D-Piece   | 66.46          | 73.36          | 79.79            | 79.77                 | 79.54                 | **77.17**             |
> | ALIT          | **70.58**      | **75.96**      | **81.55**        | **80.90**             | **79.95**             | 76.27                 |
> | KARL          | 68.09          | 73.18          | 80.64            | 80.49                 | 78.88                 | 73.41                 |
>
>
> **Evaluating on VLM (VQA)**
>
> | Approach      | VLM Score (32 tokens) | VLM Score (64 tokens) | VLM Score (256 tokens) |
> |---------------|-----------------------|------------------------|-------------------------|
> | One-D-Piece   | 62.4                  | 71.4                   | 76.7                    |
> | ALIT          | **64.1**              | **71.6**               | 77.0                    |
> | KARL          | 63.8                  | 68.4                   | **77.6**                |
>
> ---
>
> **Linear Probing Experiments:**
>
> We also conducted linear probing by training a single-layer linear classifier on top of the latent tokens produced by KARL and ALIT. Using only the first 32 1D latent tokens, ALIT achieves 44.8% Top-1 classification accuracy on ImageNet-1K, while KARL attains 41.5%. When both the 2D image tokens and the first 32 1D tokens are used, performance improves to 51.6% for ALIT and 45.0% for KARL—surpassing with the non-adaptive Titok tokenizer (48.0%).
>
> This highlights a few key observations:
> - As also noted in the ALIT paper, linear probing accuracy improves with more recurrent iterations during training—potentially due to added training FLOPs and hierarchical gradient flow across RNN layers. This may partly explain ALIT’s edge over KARL.
> - Investigating how compression or minimality impacts the linear separability of learned representations is a promising direction for future work.
> - The One-D-Piece paper reports noticeably lower linear probing accuracy (mid-30s). We suspect this may be due to an incorrect probing setup and will re-run the evaluation in our framework to report corrected numbers in the final draft.
>
>
> ---
>
> **Justification for Emphasis on Reconstruction Metrics**
>
> One minor comment, we believe reconstruction metrics still remain a valuable indicator of performance. A compact representation that enables near-lossless or low-\\( \epsilon \\)-bounded reconstruction intuitevly will retain the key semantics necessary for most downstream tasks—although *requiring different decoding capacities (exciting direction for future work)*. Moreover, single-image reconstruction metrics also echoes the principles of KC and AIT in general: identifying the shortest program that can reproduce the input data. Finally, we believe thoroughly exploring the connection between compressed representations and downstream performance is a compelling standalone research direction, and is one we are actively interested in.
>
>
> ---
>
> **Comparison with More Baselines**
>
> We compared against two recent strong baselines—ALIT (ICLR 2025) and One-D-Piece (2025). Other notable works include Flextok, ElasticTok, (non-adaptive) FlowMo, Titok, and older methods like VAE and VQGAN. FlexTok and ElasticTok fall under the Matryoshka-style tokenizers, and since ALIT, One-D-Piece, and Titok all adopt a similar 2-stage GAN-based training pipeline with 2D VQGAN/VAE tokens, we chose them for a fair, apples-to-apples comparison focusing on our core contributions.
>
> Furthermore, Flextok models are much large-scale (e.g., trained with REPA-style losses using DINO, which is trained on much broader datasets), making direct comparisons with KARL, ALIT, and One-D-Piece less fair in scope. That said, based on the reviewer’s request, we will include Flextok results in the final version.
>
> ---
>
> **Limitations**
>
> We apologize for missing the limitations section in hindsight.
>
>
> - FID gap despite strong per-image metrics: KARL performs competitively or better than existing tokenizers on all metrics except the distribution-level metric FID. This may be due to KC-guided training favoring the search for the shortest program per image—optimizing for single-image metrics (LPIPS etc)—rather than global distribution alignment. A deeper analysis of this trade-off would be valuable.
>
> - Inability of KC to distinguish noise from structure: Kolmogorov Complexity assigns high complexity to both noisy and highly structured images. Higher-order measures such as Sophistication can make this distinction and form the basis of our proposal for interestingness. However, our current interestingness formulation is a simple delta on KC, not a learned signal, also not our main focus.
>
> - General limitation of all adaptive tokenizers: Current methods, including KARL, are constrained to a maximum token budget per image (e.g., 256 tokens for 256×256 images), limiting granularity.
> - **Future Work:** Elaborate study of linear separability or required size of decoder for different downstream tasks.
> - **Future Work:** Exploring the use of learned compressed tokens to train downstream models—such as vision-language models or video models—is a promising and timely direction for further research. As acknowledged by the reviewer, this is challenging due to compute constraints in an academic setting, specially learning both a tokenizer, doing scaling laws at tokenizer levels and then training downsteam tasks all in one paper.
>
> ---
>
> Finally, we hope we have addressed all of the reviewer’s concerns. As acknowledged, the paper already includes several key contributions suitable for NeurIPS:
> (a) a novel single-pass adaptive tokenizer,
> (b) a loss-conditioned training scheme inspired by the seminal Upside-Down RL framework,
> (c.) theoretical grounding via Kolmogorov Complexity, and
> (d) thorough evaluation—both through scaling laws and now via downstream tasks (thanks to the reviewer).

---

> > ### Comment · Reviewer_2QbU · 2025-08-08
> >
> > Thanks for your detailed repsonses. I have raised my rating.

---

> > > ### Author Response · Authors · 2025-08-08
> > > **Response to Reviewer 2QbU**
> > >
> > > We appreciate you acknowledging our rebuttal—thank you!
> > > If possible, we’d be grateful for some visibility on the updated rating or any feedback, to help us understand the paper’s current standing with respect to the acceptance threshold, given that all queries have been addressed.
> > >
> > > Best Regards!

---

> ### Author Response · Authors · 2025-08-07
> **Follow up**
>
> Thank you again for your review. We believe we’ve addressed all your queries and appreciate your feedback—it has helped us make the paper more complete. As the discussion period is coming to a close, we would be grateful for any follow up comments and rating update. We remain committed to improving the draft based on any additional input.
>
> ---
>
> We’d also like to share that we have now evaluated Flextok. Despite being a single-pass method, significantly more efficient at inference, and trained only on ImageNet (while Flextok uses DINO supervision), KARL performs on par with—or better than—Flextok on single-image metrics.
>
> Additionally, we created a figure outlining the taxonomy of recent adaptive tokenizers for the latest version of our paper. However, due to NeurIPS rebuttal constraints, we couldn’t include it in the rebuttal document. We're happy to provide a textual summary here if helpful, though we've omitted it for brevity, as it's already included in the full rebuttal response.
>
> ---
>
> Finally, as acknowledged by you earlier, we believe the paper contributes meaningfully to the scientific study of visual tokenizers, with several novel and important elements. We trust that the main submission, along with the rebuttal—which addresses all raised concerns—will receive a fair and thorough evaluation.

---

### Author Response · Authors · 2025-08-06
**General Response (also awaiting reviewer's response of rebuttal)**

We thank all reviewers for their thoughtful feedback and recognition of our contributions—a novel one-shot adaptive tokenizer (2QbU, 95ih, HAzc, YZ29), insightful connections to Kolmogorov Complexity (2QbU, 95ih), solid theoretical grounding (2QbU), systematic architectural study (HAzc), and clear, well-written presentation (2QbU).

We have improved the writing, added comprehensive downstream and out-of-distribution evaluations, and addressed all reviewer concerns.

---

Through this response, we wish to share additional thoughts with the AC and reviewers on why we believe the paper is both timely and significant—positioned at the intersection of emerging trends in **adaptive tokenization** and foundational ideas from **Algorithmic Information Theory (AIT).**

### **Adaptive Tokenization**
Existing fixed-length tokenizers like VAE, VQGAN, CLIP or TiTok use fixed token budgets per image or frame, which is inefficient for high-resolution or long-horizon video. Computation should ideally scale with the information content—*echoing core principles of Algorithmic Information Theory (AIT).* Towards this goal, adaptive tokenization has rapidly gained traction, with recent works like ALiT, ElasticTok, One-D-Piece, and FlexTok advancing this emerging direction. We believe it's both timely and foundational.

A key limitation of prior adaptive tokenizers methods is their reliance on iterative inference-time search to select token counts. *KARL overcomes this with a single-pass, one-shot approach, marking a step toward practical and efficient adaptive tokenization.* In the camera-ready, we will include a visual taxonomy of recent methods to better situate KARL within this landscape.

### **Algorithmic Information Theory (AIT)**
AIT is widely regarded as a strong theoretical foundation for understanding intelligent systems. It echoes influential concepts such as Solomonoff Induction, Kolmogorov Complexity, and Simplicity Bias (Occam’s Razor). Solomonoff Induction posits that for any computable data point, the underlying algorithmic program that generates it can, in principle, be found by iteratively searching over all possible programs—prioritizing those with higher prior probability, i.e., shorter programs with lower Kolmogorov Complexity. Solomonoff Induction, and thus the centrality of Kolmogorov Complexity, has often been linked to the study of general intelligence, with several influential papers exploring this connection [5].
*AIT was inspirational for our project and KARL learns to approximate the search of the shortest program or minimal token count for an image, in a single pass.*

**Seminal works supporting our claims:** Seminal works on the Kolmogorov Structure Function and Minimum Description Length [1] explicitly frame Kolmogorov Complexity as a foundation for lossy compression—the core objective of our paper: extracting the minimal number of tokens needed to achieve a target reconstruction quality aligned with downstream task performance. For instance, Section II.4 (page 5) of [1] makes this connection explicit. The top-right paragraph of the first page further suggests that KSF—and related notions like sophistication and logical depth—can be seen as higher-order constructs derived from Kolmogorov Complexity. This perspective directly informs our learned approximations of such quantities.

**Similar works in LLMs:** In the language domain, recent studies increasingly draw parallels between LLM behavior and AIT or Solomonoff Induction [6]. Several state-of-the-art LLMs explore *adaptive chain-of-thought (CoT) lengths* (analogous to adaptive visual tokens), with a recent Anthropic paper [2] revealing an inverse scaling law—shorter CoTs often improve generalization. [3] further identifies an optimal minimal CoT length. Similarly, H-Net [4], recently proposed as a dynamic tokenization framework for text, learns where to segment raw language inputs—*analogous to how KARL learns when to stop emitting image tokens*. These developments reflect a broader trend toward learned, dynamic compression across both vision and language.

---

Overall, with all reviewer queries addressed, we believe the paper advances the scientific study of visual tokenizers by introducing a fresh perspective grounded in AIT.

We look forward to the reviewers’ response and hope the reviewers and AC will recognize the paper’s full potential. We're happy to receive further feedback and are committed to improving the paper in any way possible.

Thanks and best regards!

----

[1] Kolmogorov’s Structure Functions and Model Selection.
[2] Inverse Scaling in Test-Time Compute.
[3] When More is Less: Understanding Chain-of-Thought Length in LLMs.
[4] Dynamic Chunking for End-to-End Hierarchical Sequence Modeling.
[5] Universal Intelligence: A Definition of Machine Intelligence.
[6] Large Language Models as Computable Approximations to Solomonoff Induction.

---

### Note · Authors · 2025-08-14

Dear AC and reviewers,

We would like to highlight that **all reviewer queries have been fully addressed**, with no remaining concerns at all. All the reviewers have acknowledged this.

**Summarizing Paper Contributions:** (a) Novel single-pass adaptive tokenizer (b) Novel loss-conditioned training with parallels to the seminal Upside-Down RL framework, (c) Theoretical link to Algorithmic Information Theory and Kolmogorov Complexity (d) State-of-the-art performance on all single-image reconstruction metrics and diverse downstream tasks (classification, depth estimation, segmentation, CLIP alignment, VQA).

With all concerns addressed within the response limit, we trust the paper’s fresh scientific contributions will be valued.

Best regards

---

### Decision · Program_Chairs · 2025-09-17

**Decision:**

Accept (poster)

**Comment:**

This paper introduces KARL, a novel and efficient single-pass adaptive tokenizer that performs comparably to state-of-the-art iterative methods. The authors successfully addressed initial reviewer concerns by providing comprehensive evaluations on various downstream tasks and clarifying the method's theoretical grounding. The paper's novelty, good empirical validation, and thorough rebuttal make it a good contribution.